# Robust repression of tRNA gene transcription during stress requires protein arginine methylation

Richoo B Davis[1], Neah Likhite[1], Christopher A Jackson[1] , Tao Liu[2] , Michael C Yu[1]

Protein arginine methylation is an important means by which protein function can be regulated. In the budding yeast, this modification is catalyzed by the major protein arginine methyltransferase Hmt1. Here, we provide evidence that the Hmt1-mediated methylation of Rpc31, a subunit of RNA polymerase III, plays context-dependent roles in tRNA gene transcription: under conditions optimal for growth, it positively regulates tRNA gene transcription, and in the setting of stress, it promotes robust transcriptional repression. In the context of stress, methylation of Rpc31 allows for its optimal interaction with RNA polymerase III global repressor Maf1. Interestingly, mammalian Hmt1 homologue is able to methylate one of Rpc31's human homologue, RPC32β, but not its paralogue, RPC32α. Our data led us to propose an efficient model whereby protein arginine methylation facilitates metabolic economy and coordinates protein-synthetic capacity.

## Introduction

In eukaryotes, RNA polymerase III (Pol III) transcribes small, untranslated RNAs such as 5S rRNAs and tRNAs, highly abundant molecules that comprise ~15% of total cellular RNA by weight and are requisite elements for protein synthesis (reviewed in Geiduschek and Tocchini-Valentini (1988), Turowski and Tollervey (2016), Lesniewska and Boguta (2017)). The Pol III transcription apparatus is highly conserved among eukaryotes. In the budding yeast *Saccharomyces cerevisiae*, the holoenzyme is composed of 17 subunits, 12 of which are either shared with, or have homologs among the other RNA polymerases. Of the remaining five subunits, Rpc82, Rpc34, and Rpc31 form a Pol III-specific trimeric subcomplex that contributes to transcription initiation (Werner et al, 1992; Thuillier et al, 1995). Pol III is recruited to tRNA genes by two general transcription factors, TFIIIB and TFIIIC. The latter is a complex that recognizes sequence-specific promoter elements and guides the concerted binding of three TFIIIB subunits (TBP, Brf1, and Bdp1) to the transcription start site. The resulting TFIIIB/C complex recruits Pol III and contributes to the formation of an open promoter complex (reviewed in Graczyk et al (2018)).

Pol III-mediated transcription is robust under optimal conditions. However, under nonfavorable growth conditions or after exposure to other forms of stress, Pol III transcription is repressed by the negative regulator Maf1 (Upadhya et al, 2002). In the latter context, Maf1 is dephosphorylated and then translocates from the cytoplasm to the nucleus, where it binds to and regulates the activity of Pol III (reviewed in Boguta (2013), Willis and Moir (2018)). Cryo-EM–based analysis of the structure of a Pol III–Maf1 complex revealed that Maf1 binding leads to rearrangement of the Rpc82/34/31 subcomplex (Vannini et al, 2010) and that this inhibits the interaction between Rpc34 and TFIIIB subunit Brf1 and thus prevents recruitment of Pol III to promoters.

The Pol III machinery has been shown to undergo several types of post-translational modifications and these have been implicated in its regulation (reviewed in Chymkowitch and Enserink (2018)). Recently, a comprehensive proteomics analysis revealed that human RPC4 (yeast Rpc53) and RPC7 (yeast Rpc31) can undergo protein arginine methylation (Geoghegan et al, 2015). This modification is catalyzed by members of the protein arginine methyltransferase (PRMT) family of enzymes, which are divided into four subtypes based on the type of methylarginine formed (reviewed in Bedford and Richard (2005), Morales et al (2016), Blanc and Richard (2017)). PRMT1 is the most conserved of the type I PRMTs and it catalyzes the formation of monomethylarginine and asymmetric dimethylarginine. In yeast, Hmt1 (also termed Rmt1) is the homolog of mammalian PRMT1, and it is the only known type I PRMT in the budding yeast (Gary et al, 1996; Henry & Silver, 1996).

Our previous study revealed that Hmt1 associates with most tRNA genes in vivo and the tRNA abundance in Hmt1 loss-of-function mutants are lower than in WT (*HMT1*) counterparts (Milliman et al, 2012). However, the molecular mechanism by which Hmt1 influences tRNA abundance remained unclear. In this study, we demonstrate that the degree of association of Hmt1 with tRNA genes correlates with the transcriptional activities of the latter. We also show that Hmt1 methylates Rpc31 in vitro, that this ability is conserved in one of the two human homologs of Rpc31, and that

[1]Department of Biological Sciences, State University of New York at Buffalo, Buffalo, NY, USA  [2]Department of Biochemistry, Jacobs School of Medicine and Biomedical Sciences, State University of New York at Buffalo, Buffalo, NY, USA

Correspondence: mcyu@buffalo.edu
Tao Liu's present address is Department of Oncology, Roswell Park Comprehensive Cancer Center, Buffalo, NY, USA

under optimal growth conditions, Rpc31 methylation promotes biogenesis of precursor tRNAs (pre-tRNAs), but in the setting of stress, it represses the biogenesis of these pre-tRNAs. Our finding further suggests that the observed difference in outcomes is due to Rpc31 methylation affecting with the interaction between Pol III and its repressor, Maf1.

## Results

### Hmt1 occupancy at tRNA genes correlates with RNA Pol III transcriptional activity

In vivo, Hmt1 is associated with many tRNA genes, and in both hmt1Δ and Hmt1-G68R (a catalytically inactive mutant of Hmt1) cells grown in rich medium, tRNA abundance is higher than that in their HMT1 counterparts (Milliman et al, 2012). These observations led us to investigate the relationship between the transcription of tRNA genes and the association of Hmt1 with these loci. The transcriptional activity of tRNA genes is high in rich medium and low in the contexts of nutrient deprivation (Clarke et al, 1996), treatment with the antifungal compound chlorpromazine (CPZ) (Upadhya et al, 2002), and treatment with tunicamycin (Li et al, 2000). We subjected exponentially growing cultures of yeast cells expressing Myc-tagged Hmt1 to each of these conditions and used chromatin immunoprecipitation (ChIP) to measure the extent to which Hmt1 associated with tRNA genes before and after treatment (Fig 1). This

analysis revealed reductions in the association of Hmt1 with these tRNA genes upon nutrient deprivation (Fig 1A), treatment with CPZ (Fig 1B), or treatment with tunicamycin (Fig 1C). In particular, the reduction in association after nutrient deprivation is similar to the reduction in occupancy seen for the RNA Pol III machinery under the same conditions (Roberts et al, 2003). We note that Hmt1 occupancy in untreated samples across these treatments had varied level of occupancy and this is likely attributed to technical differences in the immunoprecipitation efficiency for each immunoprecipitation, as the background signals seen with the negative control genes were also slightly higher as well. Nevertheless, the trend we observed for each of the three different treatments reflects a decreased association of Hmt1 with tRNA genes that correlates with the levels of transcription of the latter.

### Hmt1 methylates the Pol III subunit Rpc31 in vitro

The RNA hybridization data for Hmt1-G68R hinted that a substrate of this methyltransferase may be crucial for Pol III transcription (Milliman et al, 2012). In hmt1Δ cells, the loss of methylation of such a substrate is likely responsible for the observed changes in tRNA levels. Examination of the amino acid sequences of all Pol III subunits for the presence of RGG tripeptides or RG repeats, which are common methylation motifs in substrates of Hmt1, identified Rpc31 as a candidate (Fig 2A). Notably, this protein had previously been suggested as a putative Hmt1 substrate based on another in silico analysis (Frankel & Clarke, 1999).

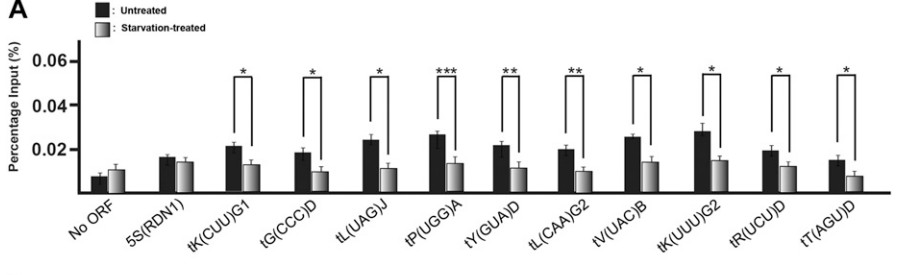

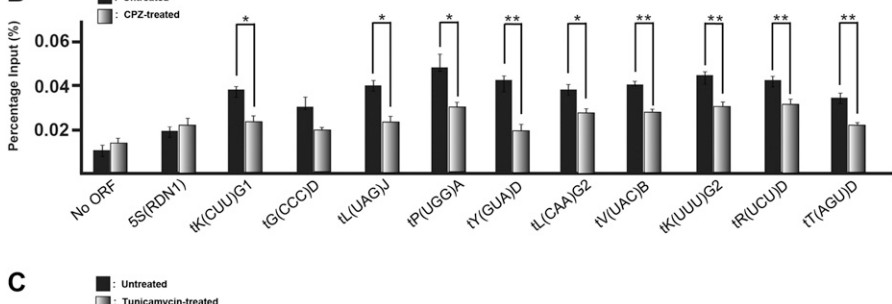

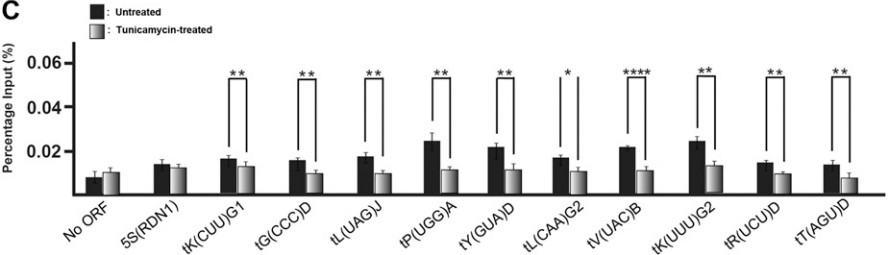

**Figure 1. Hmt1 occupancy at tRNA genes is decreased under stress conditions.**
**(A–C)** Hmt1 occupancy across tRNA genes was measured in yeast cells before and after nutrient deprivation (A), treatment with CPZ (B), or treatment with tunicamycin (C). qPCR results for products of ChIP are displayed in bar graphs. Percentage of input is calculated as $\Delta C_T$, with error bars representing the SEM of three biological samples (n = 3). P-value as calculated by t test: *<0.05; **<0.01; and ***<0.001.

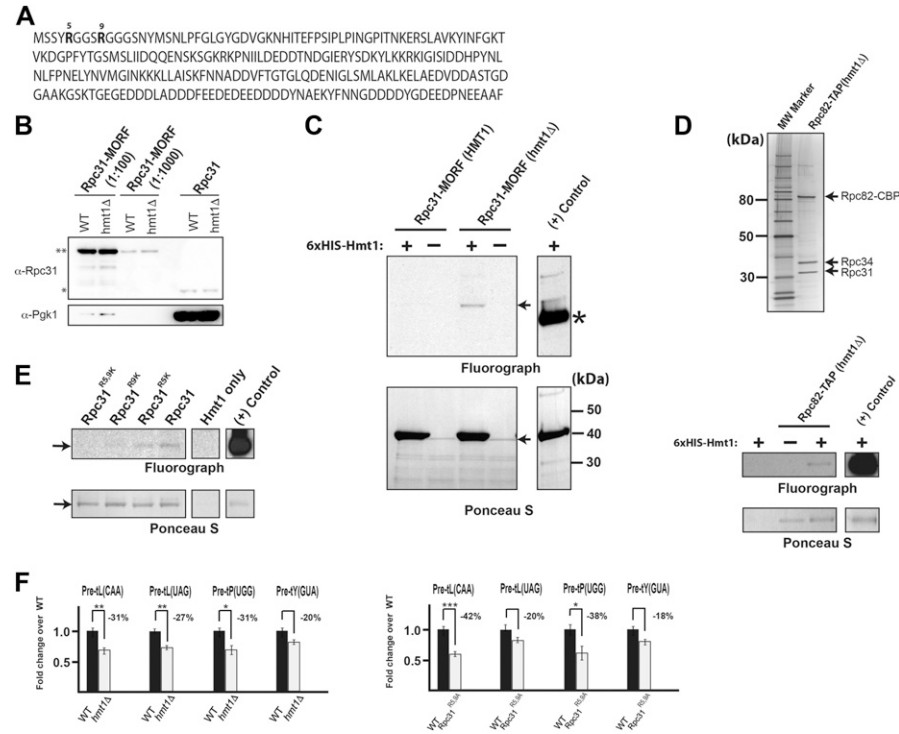

**Figure 2. Under optimal growth conditions, Hmt1 methylates Rpc31 and loss of this modification adversely affects biogenesis of pre-tRNAs.**
**(A)** The amino acid sequence of yeast Rpc31 with methylated arginines at positions 5 and 9 denoted in bold lettering. **(B)** Immunoblotting showing the relative levels of Rpc31-MORF to endogenous Rpc31 was analyzed using lysates made from WT and *hmt1Δ* cells. Double asterisks denote MORF-tagged Rpc31 and a single asterisk denotes endogenous Rpc31. The level of Pgk1 is used as a loading control for the relative total protein levels loaded. **(C)** In vitro methylation of Rpc31 from the yeast MORF collection, after purification from WT and *hmt1Δ* cells, using recombinant Hmt1 and [methyl-$^3$H]-SAM. The full protein complement in each reaction was resolved on a 4–12% SDS–PAGE; methylation was visualized by fluorography (arrow) and protein levels by Ponceau S staining. Recombinant GST-tagged Rps2 served as a positive control (highlighted by asterisk). **(D)** Biochemical purification of TAP-tagged Rpc82 from *hmt1Δ* cells. TAP was performed from *hmt1Δ* cells expressing TAP-tagged Rpc82 (top panel). The purified proteins were resolved on a 4–12% SDS–PAGE and the gel was silver stained to determine the protein composition (bottom panel). The TAP-purified Rpc82 and associated proteins were subjected to an in vitro methylation assay using recombinant Hmt1 and [methyl-$^3$H]-SAM. The full protein complement in each reaction was resolved on a 4–12% gel by SDS–PAGE. Methylation of Rpc31 was visualized by fluorography and protein levels by Ponceau S staining. Recombinant GST-tagged Rps2 served as a control. **(E)** In vitro methylation of GST-tagged WT Rpc31, Rpc31$^{R5K}$, Rpc31$^{R9K}$, and Rpc31$^{R5,9K}$, after purification from *Escherichia coli*, using recombinant Hmt1 and [methyl-$^3$H]-SAM. Visualization of methylation and protein as in Part (B). Recombinant GST-tagged Rps2 served as a positive control. **(F)** Fold change in levels of pre-tRNAs in *hmt1Δ* or Rpc31$^{R5,9A}$ versus WT cells under optimal growth condition in either SC + glucose (for WT versus *hmt1Δ* cells) or YPD (for WT versus Rpc31$^{R5,9A}$ cells), as assessed by hybridization of probes to intronic regions. Bars show abundance of pre-tRNAs tL(CAA), tL(UAG), tP(UGG), and tY(GUA) in *hmt1Δ* (left panel) or Rpc31$^{R5,9A}$ (right panel) relative to those in WT cells. In each case, the signal was normalized based on the levels of U4 snRNA. Error bars represent the SEM of three biological replicates (n = 3). *P*-value as calculated by *t* test: *<0.05 and **<0.01.

To test whether Rpc31 can be methylated by Hmt1, we used the yeast moveable ORF (MORF) expression plasmid collection (Gelperin et al, 2005) to overexpress and purify C-terminal epitope-tagged Rpc31 in the presence of endogenous Rpc31 expression. The rationale for choosing this approach is because C-terminal tagging of endogenous Rpc31 has proven to be detrimental for cell growth and N-terminal tagging of Rpc31 is likely to interfere with the predicted methylation motif located close to the N-terminal end of the protein. The level of MORF-tagged Rpc31 is much more in abundance than endogenous Rpc31, based on our immunoblot analysis (Fig 2B). Using C-terminal epitope-tagged Rpc31 purified from *HMT1* and *hmt1Δ* cells, we set up an in vitro methylation assay, and our results revealed a signal corresponding to methylation of Rpc31 in only the Rpc31 purified from *hmt1Δ* cells (Fig 2C, compare position of asterisk on the fluorograph in the *HMT1* and *hmt1Δ* lanes). This is consistent with Rpc31 purified from *hmt1Δ* cells being in the hypomethylated form and thus serving as a good substrate for in vitro methylation but Rpc31 purified from *HMT1* cells being at least partially (and potentially fully) methylated and thus a poorer substrate for methylation in vitro.

To demonstrate the physiological relevance of Rpc31 methylation, we carried out a tandem-affinity purification (TAP) of TAP-tagged Rpc82 that was expressed in *hmt1Δ* cells. Our purification results revealed Rpc34 and Rpc31 that were co-purified with TAP-tagged

Rpc82 in corresponding stoichiometry (Fig 2D, top). When we subjected the entire TAP-purified fractions to an in vitro methylation assay, a signal that corresponds to the co-purified Rpc31 was detected (Fig 2D, bottom). Thus, endogenous Rpc31 can be methylated when present as a complex along with Rpc34 and Rpc82, which resembles a more physiological relevant condition that exists in a cell. Taken together, our data collectively demonstrated that Rpc31 is an in vitro substrate of Hmt1 and is likely to be methylated by Hmt1 under physiological conditions.

### Hmt1 methylates the Rpc31 arginines at positions 5 and 9

To determine how arginine methylation influences Rpc31 function, it was necessary to identify the residues that are modified by Hmt1. Of the six arginines present in Rpc31, only those in positions 5 and 9 are located within the N-terminal RGG motif (Fig 2A). To test the methylation potential of these two arginines, we mutated them to lysines individually and in combination (Rpc31$^{R5K}$, Rpc31$^{R9K}$, or Rpc31$^{R5,9K}$) and subjected recombinant Rpc31 harboring these substitutions to in vitro methylation (Fig 2E). Our data revealed a reduction in the methylation signal for both Rpc31$^{R5K}$ and Rpc31$^{R9K}$ (Fig 2E, compare the signal from Rpc31 to Rpc31$^{R5K}$ or Rpc31$^{R9K}$ lanes). Moreover, in the Rpc31$^{R5,9K}$ double mutant, this signal was completely abolished (Fig 2E, compare the signal from Rpc31 lane to

Rpc31[R5,9K] lane). Although each substitution had an impact on the total methylation of Rpc31, it was clear that both arginines are methylated.

### In both *hmt1Δ* and Rpc31[R5,9A] mutants, the biogenesis of pre-tRNAs is impaired when cells are maintained under optimal growth conditions

Our previous conclusions regarding RNA hybridization data for *hmt1Δ* cells maintained under optimal growth conditions, that is, that tRNA levels increased in this context, were based on normalization to the Pol I transcript 5.8S (Milliman et al, 2012). We have since changed to using levels of the U4 snRNA transcript for normalization, for the following reasons. First, Pol I and III are extensively co-regulated (Li et al, 2000; Briand et al, 2001), making a Pol II transcript a potentially better means of assessing the impact of Hmt1 on the transcriptional activities of Pol III. Second, the U4 snRNA, in particular, has been used in normalizing the results of RNA hybridization studies investigating transcription by Pol III (Sethy-Coraci et al, 1998; Li et al, 2000). Third, the loss of Hmt1 or its catalytic activity does not alter RNA Pol II–mediated gene transcription (Yu et al, 2004).

We repeated our RNA hybridization assay, focusing on tRNAs that were bound by Hmt1 and also contain introns. The presence of an intron on such tRNA allows us to follow the fate of the precursors by using a probe that binds to tRNA introns. This allows us to detect the earliest intermediate resulting from the transcription of these tRNA genes. When we normalized our data using U4, we found the levels of pre-tRNAs to be lower in the *hmt1Δ* versus WT cells grown in both YPD and synthetic complete (SC) media supplemented with glucose (SC + glu). However, the differences seen in cells grown in YPD were smaller or not as consistent when compared with SC + glu grown cells. When *hmt1Δ* cells were grown in SC + glu, we consistently see a decrease in the levels of pre-tRNAs to be lower in the *hmt1Δ* cells across all four pre-tRNAs tested when compared with the WT cells (Fig 2F, left panel). This is in contrast to our previous conclusions, based on normalization of the data to the levels of the 5.8S transcript.

To determine whether methylation of Rpc31 is responsible for this change, we repeated the same experiment using a Rpc31[R5,9A] mutant grown in YPD. Results from our RNA hybridization data demonstrated a consistent decrease in the levels of pre-tRNAs across the same four pre-tRNAs tested (Fig 2F, right panel). Together, these findings suggest that Hmt1 is required to promote transcription of the tested tRNA genes under optimal growth conditions and that it does so by methylating Rpc31.

### Investigating repression of tRNA biogenesis under stress in the *hmt1Δ* and Rpc31[R5,9A] mutant strains

In the context of stress, Pol III transcription is robustly repressed; this ensures cell survival by promoting metabolic economy (reviewed in Warner (1999)). We examined the extent to which Hmt1 contributes to this process by investigating the ability of the *hmt1Δ* and Rpc31[R5,9A] mutants to repress tRNA biogenesis in the context of stress. For the four tRNA species examined in Fig 2F, we carried out RNA hybridizations using total RNA extracted from cells grown in YPD before and after treatment by CPZ. For comparison, we calculated the percentage of pre-tRNA abundance for each strain and treatment and compared these values with the one calculated from untreated WT cells. To ensure that our RNA hybridization results accurately reflects the levels of pre-tRNA examined, we used two internal normalizing controls, U3 and U5, in addition to U4. Both U3 and U5 have been used previously as internal controls for normalization of tRNA abundance (Moir et al, 2006; Arimbasseri et al, 2016). In the four pre-tRNAs examined, we saw that the loss of Hmt1 compromised the cell's ability to robustly repress pre-tRNA biogenesis in the context of stress (Figs 3A and S1A). This trend is also true for the Rpc31[R5,9A] mutants where diminished repression of pre-tRNAs was observed in the mutants versus those seen in WT cells (Figs 3B and S1B). Although there are slight variations depending on the use of U3, U4, or U5 as normalizing controls, the overall trend in which tRNA repression is less robust in either *hmt1Δ* or Rpc31[R5,9A] mutants remains consistent.

To better test the differences among the three strains for significance, we carried out Tukey's HSD on U4-normalized RNA hybridization data. The results of this analysis is plotted as a violin plot (Fig 3C) where the median point within the cluster of tRNA genes tested in both *hmt1Δ* and Rpc31[R5,9A] mutants is lower than the one in the WT strain, indicating that both mutants have a lower fold decrease of pre-tRNA abundance in their treated versus untreated samples. In other words, this plot imply an overall diminished repression in *hmt1Δ* and Rpc31[R5,9A] mutants (Fig 3C). Hence, Hmt1-mediated methylation of Rpc31 contributes to the robust repression of pre-tRNA biogenesis in the context of stress, playing a role distinct from that under optimal growth conditions.

Next, we set out to determine whether the attenuated repression of tRNA biogenesis observed in *hmt1Δ* and Rpc31[R5,9A] mutants results from a change in Pol III occupancy at the relevant promoters. Using ChIP, we measured the in vivo occupancy of the same four tRNA genes by the Rpc82 and Rpc160 subunits of Pol III before and after treatment with CPZ. In both *hmt1Δ* and Rpc31[R5,9A] point mutants under stress, Rpc82 and Rpc160 occupancy at these tRNA genes are higher when compared with their levels seen in WT cells (Fig 4A and B, compare the percentage of change in WT lanes with those in *hmt1Δ* and Rpc31[R5,9A] lanes). In addition, we carried out Tukey's HSD on ChIP data from both Rpc82 and Rpc160 to examine for significance with respect to their occupancy among the three strains. The resulting violin plot (Fig 4C) indicates the median point within the cluster of Rpc82 and Rpc160 occupancy across the tested tRNA genes in both *hmt1Δ* and Rpc31[R5,9A] strains is lower than the WT, in the context of fold decrease between treated and untreated samples. What this result implies is that, under CPZ treatment, a higher Rpc82 and Rpc160 occupancy in *hmt1Δ* and Rpc31[R5,9A] mutants is observed when compared with the WT (Fig 4C). It is likely that in the context of stress, a higher occupancy of Pol III in *hmt1Δ* and Rpc31[R5,9A] mutants is responsible for the higher levels of pre-tRNAs observed in these mutants.

### Arginine methylation of Rpc31 is important for its association with Maf1 in the context of stress

Arginine methylation plays a key role in mediating proper protein–protein interactions (Bedford et al, 2000; Côté & Richard,

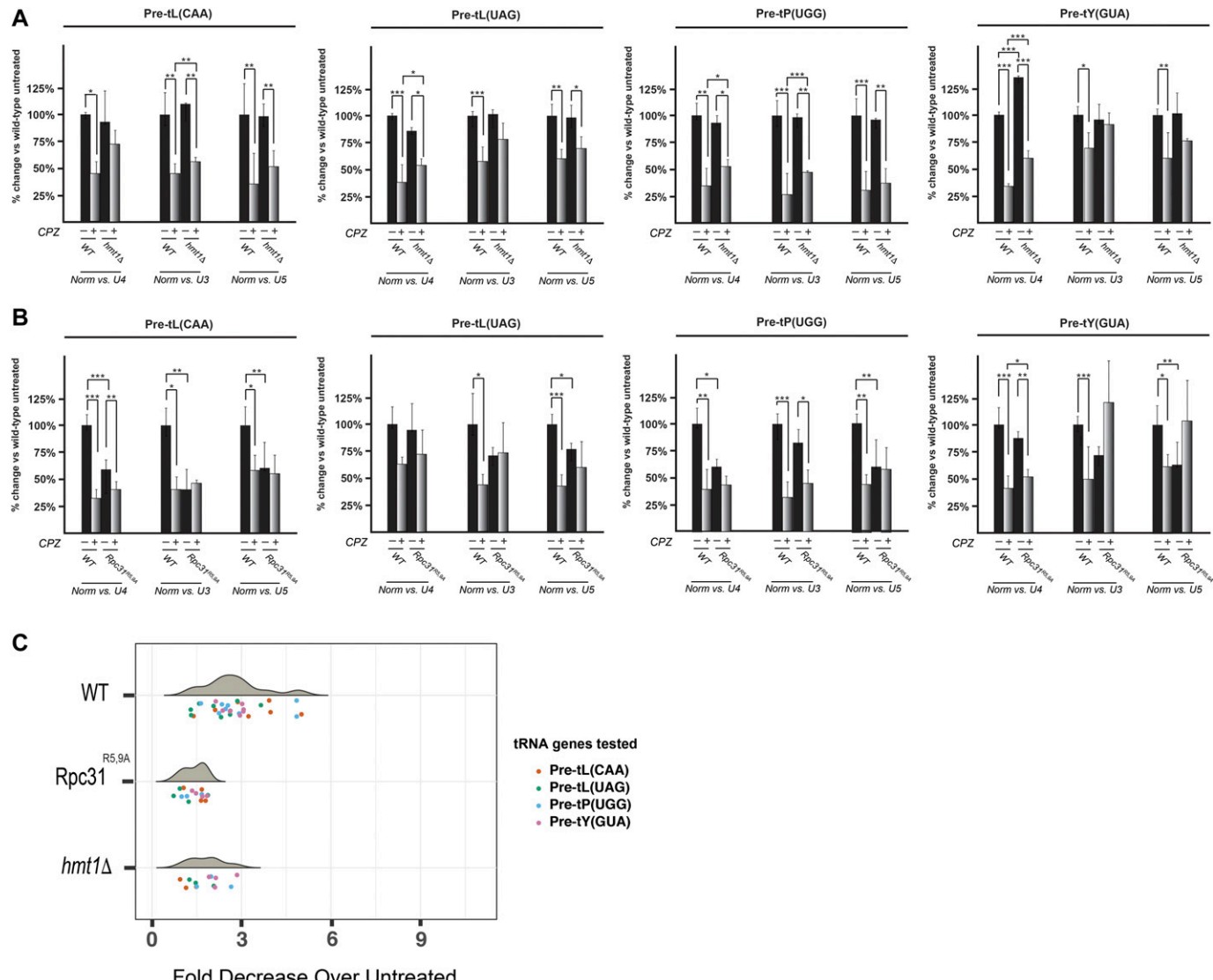

**Figure 3. In the context of stress, methylation of Rpc31 is required for robust repression of pre-tRNA biogenesis.**
**(A, B)** RNA hybridization was carried out for four pre-tRNAs in WT versus *hmt1Δ* cells (A) or WT versus Rpc31[R5,9A] cells (B) grown in YPD before and after treatment with CPZ. Ratios of signal intensities for each pre-tRNA were individually normalized against three internal controls: U4, U3, and U5. The normalized signals were plotted on a bar graph to compare against signal obtained from the untreated WT cells, which is set to 100%. Error bars represent the SEM of three biological replicates (n = 3). *P*-value as calculated by *t* test: *<0.05, **<0.01, and ***<0.001. **(C)** Fold decrease in expression of four candidate pre-tRNAs in WT, *hmt1Δ*, or Rpc31[R5,9A] cells after treatment with CPZ, as assessed by hybridization of probes to intronic regions. Signal was normalized to levels of the U4 snRNA. Analysis of variance (ANOVA) revealed significant variation among the strains (*P*-value = $1.4 \times 10^{-6}$). Post hoc Tukey's Honest significant differences method revealed a significant difference between WT and *hmt1Δ* (after adjustment for the multiple comparisons, the adjusted *P*-value is 0.0014), WT and Rpc31[R5,9A] (adjusted *P*-value is $2.8 \times 10^{-6}$). n = at least three per pre-tRNA.

2005; Cheng et al, 2007), and structural–functional studies have suggested that the recruitment of Pol III to target genes is inhibited by Maf1-mediated rearrangements of the Rpc82/34/31 subcomplex (Vannini et al, 2010). Thus, we hypothesized that Rpc31 methylation plays a key role in the biochemical association of Pol III with Maf1 during stress. We tested this hypothesis by using α-Rpc31 antibodies to immunoprecipitate proteins from yeast lysates prepared from cells expressing a Myc-tagged Maf1-7SA (Fig 5A and B). Maf1-7SA is a functional mutant that cannot be phosphorylated by PKA/Sch9 (Huber et al, 2009). It is constitutively localized in the nucleus (Lee et al, 2009) and, as such, provides an enhanced readout for our

co-immunoprecipitation (CoIP) studies. After treatment with CPZ, the association of Maf1-7SA with Rpc31 in *hmt1Δ* cells was ~40% lower than that in their WT counterparts (Fig 5A). The same effect was observed with the Rpc31[R5,9A] mutant (Fig 5B). Our CoIP likely pulled down the entire Pol III complex, as we were able to detect both Rpc34 and Rpc160 in our co-immunoprecipitates (Fig 5C). To determine whether Hmt1 physically associates with the RNA Pol III, we carried out a CoIP experiment using α-Rpc31 antibodies. We were able to capture a weak but clear association between Hmt1 and RNA Pol III complex (Fig 5D). The level of captured Hmt1 association with RNA Pol III may be reflective of Hmt1's role as an

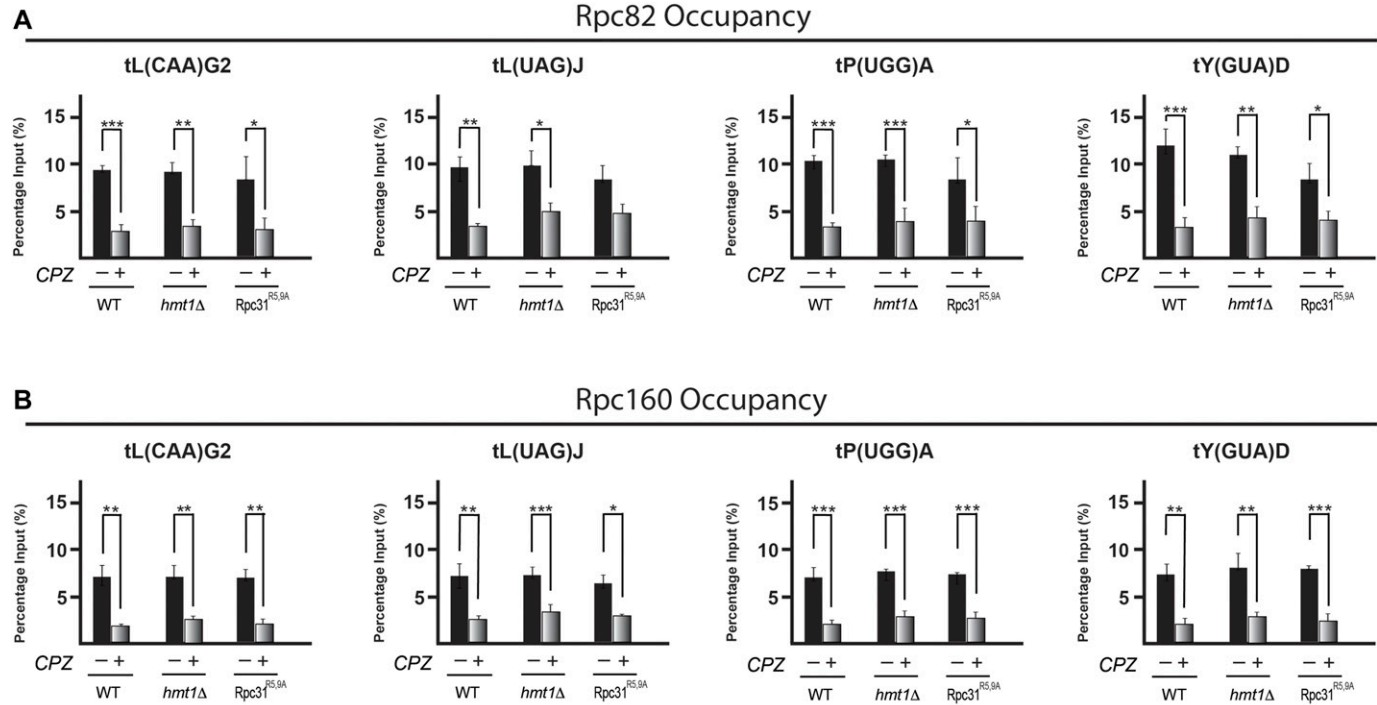

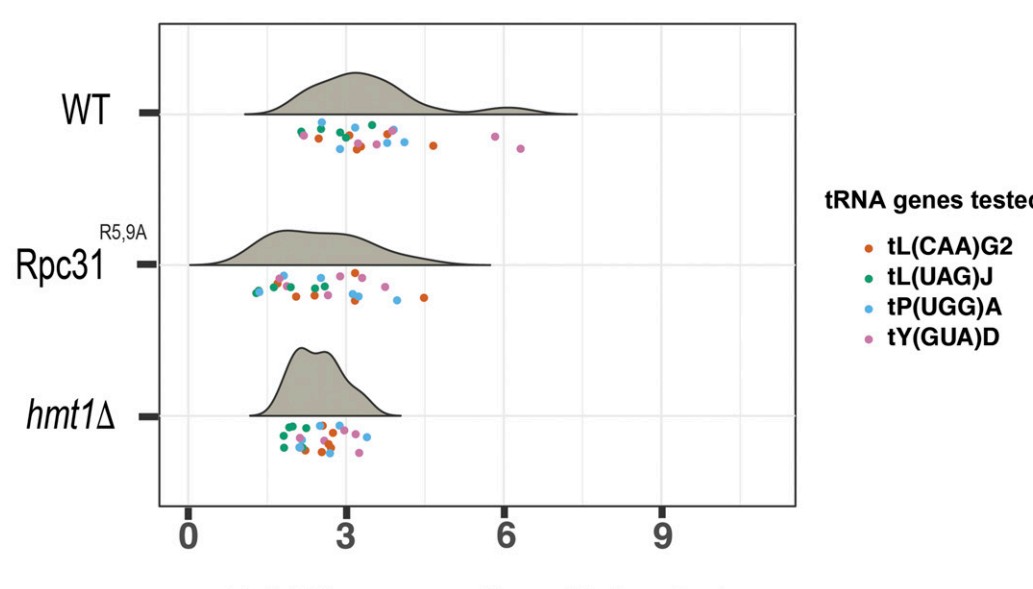

**Figure 4.  Increased RNA Pol III occupancy at tRNA genes is observed in *hmt1Δ* and Rpc31^R5,9A cells under stress.**
**(A, B)** The in vivo occupancy across the four tRNA genes for Rpc82 (A) and Rpc160 (B) was determined by ChIP. qPCR results for products of ChIP performed on WT, *hmt1Δ*, or Rpc31^R5,9A cells before and after treatment of CPZ are displayed as bar graphs. Percentage of input is calculated by $\Delta C_T$. The error bars representing SEM of three biological samples (n = 3). *P*-value as calculated by *t* test: *<0.05; **<0.01, and ***<0.001. **(C)** Fold decrease in Rpc82 and Rpc160 occupancy for four candidate tRNA genes in WT, *hmt1Δ*, or Rpc31^R5,9A cells after treatment with CPZ. qPCR was performed for products of ChIP on WT, *hmt1Δ*, or Rpc31^R5,9A cells before and after treatment with CPZ. Percentage of input is calculated as $\Delta C_T$. ANOVA on these values yielded significant variation among the three strains (*P*-value = 0.0015). Post hoc Tukey test revealed significant differences between WT and *hmt1Δ* cells (adjusted *P*-value is $5.4 \times 10^{-4}$), and between WT and Rpc31^R5,9A cells (adjusted *P*-value is $9.8 \times 10^{-4}$). n = 3 per tRNA gene.

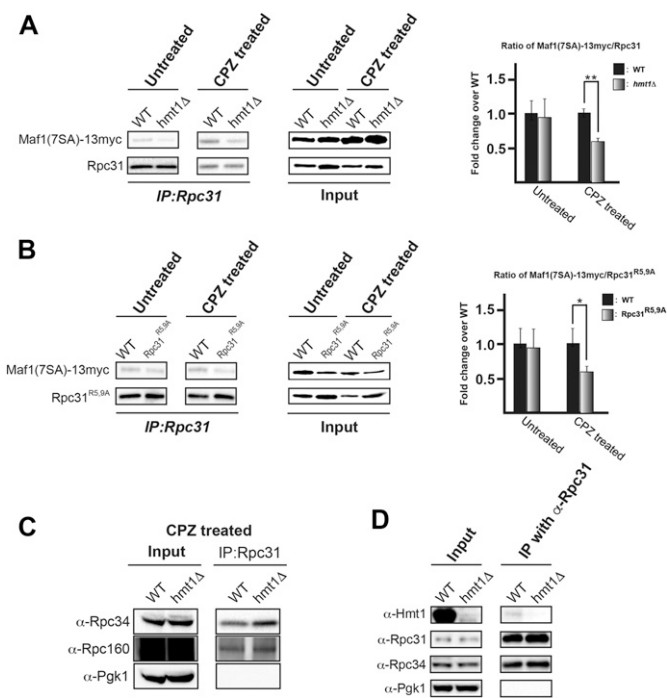

**Figure 5. During stress, methylation of Rpc31 is required for the association of Pol III with its negative regulator Maf1.**
**(A)** Rpc31 levels in yeast lysates generated from WT or *hmt1Δ* cells before and after treatment with CPZ, assessed by CoIP with an α-Rpc31 antibody. The levels of Myc-tagged Maf1-7SA were probed using an α-Myc antibody and the results are displayed in a bar graph. Error bars represent the SEM of three biological replicates (n = 3). *P*-value as calculated by *t* test: *<0.05 and **<0.01. **(B)** Rpc31 and Myc-tagged Maf1-7SA levels in yeast lysates generated from WT or Rpc31$^{R5,9A}$ cells before and after treatment of CPZ, assessed using CoIP as in (A) (including the number of replicates). *P*-value as calculated by *t* test: *<0.05 and **<0.01. **(C)** Rpc160 and Rpc34 levels in complexes immunoprecipitated with α-Rpc31 antibody. The samples were probed with α-Rpc160, α-Rpc34, and α-Pgk1 (as a negative control). **(D)** Hmt1 physically interacts with Rpc31-containing complex. CoIP of Rpc31 was carried out using cell lysates generated from WT or *hmt1Δ* cells and resolved on a 4–12% SDS–PAGE followed by immunoblotting using α-Hmt1, α-Rpc31, and α-Rpc34 antibodies to determine the levels of Hmt1, Rpc31, and Rpc34 present in the co-immunoprecipitates. The level of Pgk1 was used as a negative control in the immunoprecipitation experiment.
Source data are available for this figure.

enzyme, where its interaction with a substrate may be too transient to be completely captured. Overall, our data led us to infer that arginine methylation of Rpc31 promotes the interaction between Pol III and Maf1 and that this association is key to achieving robust repression of tRNA gene transcription in the context of stress.

A genome-wide study had previously revealed that Maf1 associates with all Pol III loci in a regulated manner, and that this association is enhanced under conditions of Pol III repression (Roberts et al, 2006). To determine whether this association is changed in *hmt1Δ* and Rpc31$^{R5,9A}$ mutants, we performed ChIP on the four tRNA genes tested above using a Myc-tagged Maf1 (Fig 6A). In both *hmt1Δ* and Rpc31$^{R5,9A}$ mutants, we observed a curtailed increase in Maf1 occupancy across the tRNA genes after the CPZ treatment when compared with the WT cells (Fig 6A, compare the percentage input in *hmt1Δ* and Rpc31$^{R5,9A}$ lanes with WT lanes). To determine the significance of Maf1 occupancy change within these

three strains, we carried out Tukey's HSD on Maf1's ChIP data in a similar manner as above. The resulting violin plot (Fig 6B) supports the notion that Maf1 occupancy across the tested tRNA genes displays a decreased median point in both *hmt1Δ* and Rpc31$^{R5,9A}$ mutants, in the context of fold increase between treated versus untreated samples. The results from this analysis implied a lower Maf1 occupancy in *hmt1Δ* and Rpc31$^{R5,9A}$ mutants in the context of stress when compared with the WT cells (Fig 6B). Taken together with the ChIP data obtained for Rpc82 and Rpc160, our observation of a reduced Maf1 occupancy in the *hmt1Δ* and Rpc31$^{R5,9A}$ mutants lend support to the notion that Maf1 is not able to fully repress the transcription of tRNA genes in these cells upon exposure to stress. This is likely due to the inability of Maf1 to fully engage in its association with a nonmethylated Rpc31 based on findings from our biochemical data above.

### The human ortholog of yeast Rpc31, RPC32β, is methylated by PRMT1 in vitro

The yeast ortholog of Rpc31/34/82 in humans is RPC32/39/62 (Wang & Roeder, 1997; Hu et al, 2002). Of all the subunits of vertebrate RNA Pol III, RPC32 is the only one for which two paralogs exist: RPC32α and RPC32β (Haurie et al, 2010). An alignment of the amino acid sequence of the Rpc31 N terminus in orthologs from various eukaryotic species indicates that its methylarginines are conserved in RPC32β, but not RPC32α (Fig 7A). To determine whether arginine methylation of Rpc31 is conserved, we performed an in vitro methylation assay, using rat PRMT1 to methylate both RPC32α and RPC32β (Fig 7B). A signal corresponding to methylation was observed for RPC32β, but not RPC32α, in the presence of rat PRMT1 (Fig 7B). To determine whether either or both of the conserved arginines on RPC32β are methylated, we generated arginine-to-lysine substitutions in RPC32β to mimic the residues in Rpc31. We observed a significant reduction in the methylation signal from both the RPC32β$^{R4K}$ and RPC32β$^{R8K}$ mutants (Fig 7B). As in the case of the Rpc31$^{R5,9K}$ double mutant, the methylation signal was completely abolished in the RPC32β$^{R4,8K}$ double mutant (Fig 7B). Overall, our data demonstrated that the arginine methylation of Rpc31 is conserved in RPC32β, but not in RPC32α.

## Discussion

### The roles of arginine methylation of Rpc31 in controlling Pol III transcription is dynamic

Determining how arginine methylation is regulated remains the greatest challenge in the field because it requires the existence of enzymes capable of demethylating methylarginines (reviewed in Yang and Bedford (2012), Wesche et al (2017)). There is no evidence that budding yeast has an enzyme that directly removes the methyl moieties from arginines, casting doubt on the reversibility of this modification in yeast. However, one can argue that if methylation of a specific residue on a substrate can allow for distinct outcomes in a biological process (e.g., negative and

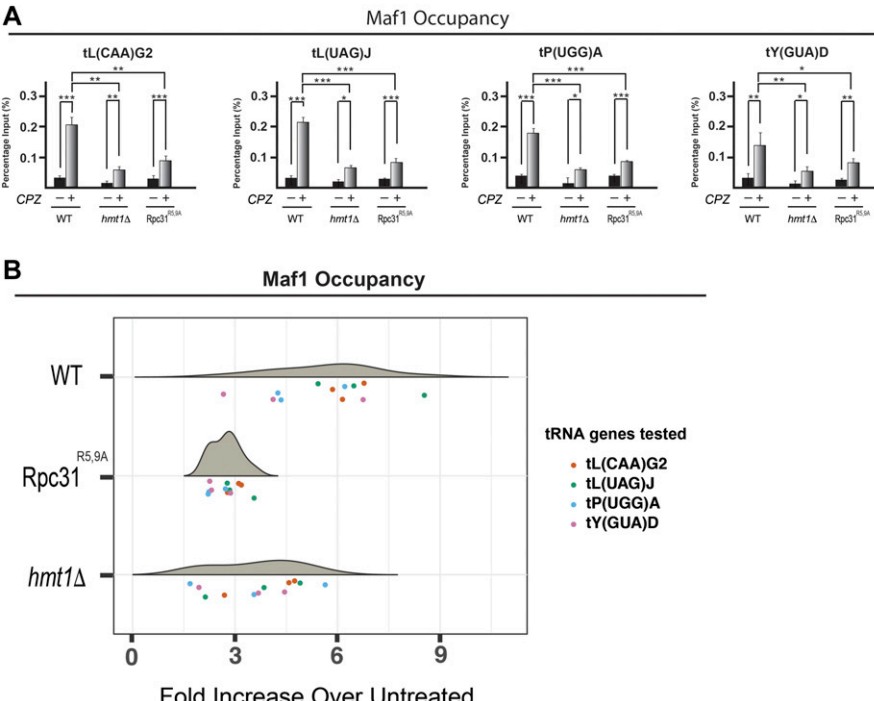

**Figure 6. Decreased Maf1 occupancy at tRNA genes is observed in *hmt1Δ* and Rpc31[R5,9A] cells under stress.**
**(A)** Maf1 occupancy across the four tRNA genes was determined by ChIP. qPCR results for products of ChIP performed on WT, *hmt1Δ*, or Rpc31[R5,9A] cells before and after treatment of CPZ are displayed as bar graphs. Percentage of input is calculated by $\Delta C_T$. The error bars representing SEM of three biological samples (n = 3). *P*-value as calculated by *t* test: *<0.05; **<0.01, and ***<0.001. **(B)** Fold increase in Maf1 occupancy of four candidate tRNA genes in WT, *hmt1Δ*, or Rpc31[R5,9A] cells after treatment with CPZ. qPCR was performed on products of ChIP in WT, *hmt1Δ*, or Rpc31[R5,9A] cells before and after treatment with CPZ. Percentage of input was calculated as $\Delta C_T$. ANOVA on these values revealed significant variation among the three strains (*P*-value = $4.6 \times 10^{-6}$). Post hoc Tukey test revealed significant difference between WT and *hmt1Δ* (adjusted *P*-value is $8.6 \times 10^{-4}$), and between WT and Rpc31[R5,9A] (adjusted *P*-value is $3.7 \times 10^{-6}$). n = 3 per tRNA gene.

positive regulation of Pol III transcription) in which the role of methylation is dependent on a specific environmental trigger (e.g., flux in the nutrient or stress), a lack-of-turnover mechanism may actually be better in achieving a higher efficiency and specificity in tuning such biological process. In addition, arginine methylation has a substantial metabolic cost (12 ATPs are required), and thus a rapid reversal of this modification is not energetically favorable (Gary & Clarke, 1998). Given this rationale, it is tantalizing to speculate that an alternative mechanism bypassing the need for a demethylase might be involved, with the same modification able to achieve distinct outcomes in the same process due to different biological signals sensed. Our data show that in cells responding to stress, the methyl marks on Rpc31 lead to robust repression of Pol III, and that they do so by facilitating proper binding between Pol III and Maf1, but that under optimal growth conditions, they ensure high-level transcription by Pol III. Although the underlying molecular mechanism is not yet clear, these marks could potentially influence how Pol III interacts with transcription factors at the pre-initiation complex. Given that Maf1 is present in the cytoplasm in cells undergoing optimal growth, it would not interfere with the interaction between methylated Rpc31 and transcription factors in this context. Thus, it is possible that Rpc31 methylation plays both positive and negative roles in Pol III transcription by participating in distinct biochemical interactions. Given the strong conservation of arginine-methylated substrates in yeast and higher eukaryotes, it is possible that a similar scenario explains the lack of a bona fide demethylase in the latter. Further research is needed to unravel the functional roles of individual methyl marks in these substrates and to better understand the prevalence of such scenarios in yeast and higher eukaryotes.

## Implication of Hmt1 recruitment on the conformational changes of RNA Pol III at tRNA genes

We have previously identified a physical association between Hmt1 and Bdp1 (Milliman et al, 2012), a subunit of TFIIIB. Our present work further demonstrates a physical association between Hmt1 and RNA Pol III, and that Hmt1 methylates Rpc31, a subunit of RNA Pol III. Together, these observations suggest that Bdp1 is instrumental in recruiting Hmt1 to the tRNA genes to allow for methylation of Rpc31 by Hmt1. Together with Rpc82 and Rpc34, this heterotrimeric subcomplex is key for promoter opening (Brun et al, 1997). Structural studies of RNA Pol III transcription initiation shows that upon TFIIIB binding, structural rearrangements leads to a shift of the Rpc82/34/31 subcomplex towards the cleft and this subcomplex undergoes further structural rearrangement that result in the stabilization of the C-terminal segment of Rpc34 and Rpc31 (Abascal-Palacios et al, 2018). Rpc31 is predicted to be disordered in the elongating Pol III structures (Hoffmann et al, 2015). Upon the formation of RNA Pol III open complex, the "stalk bridge" of Rpc31 is able to fold into a α-helix that directs this bridge to contact the RNA Pol III stalk (Abascal-Palacios et al, 2018). This contact promotes the "stalk bridge" of Rpc31 to be locked at a defined angle within the Rpc82/34/31 subcomplex (Abascal-Palacios et al, 2018). It is possible that methylation of Rpc31 plays a key role in this process by modulating the degree of disorder within Rpc31. Recent work has shown that arginine methylation has emerged as a key posttranslational modification can modulate low complexity domains within the disordered regions in a protein, thereby fine-tune the degree of interactions with other molecular partners (reviewed in Chong et al (2018), Hofweber and Dormann (2019)). Thus, methylation of Rpc31 may provide a similar role in this instance to change

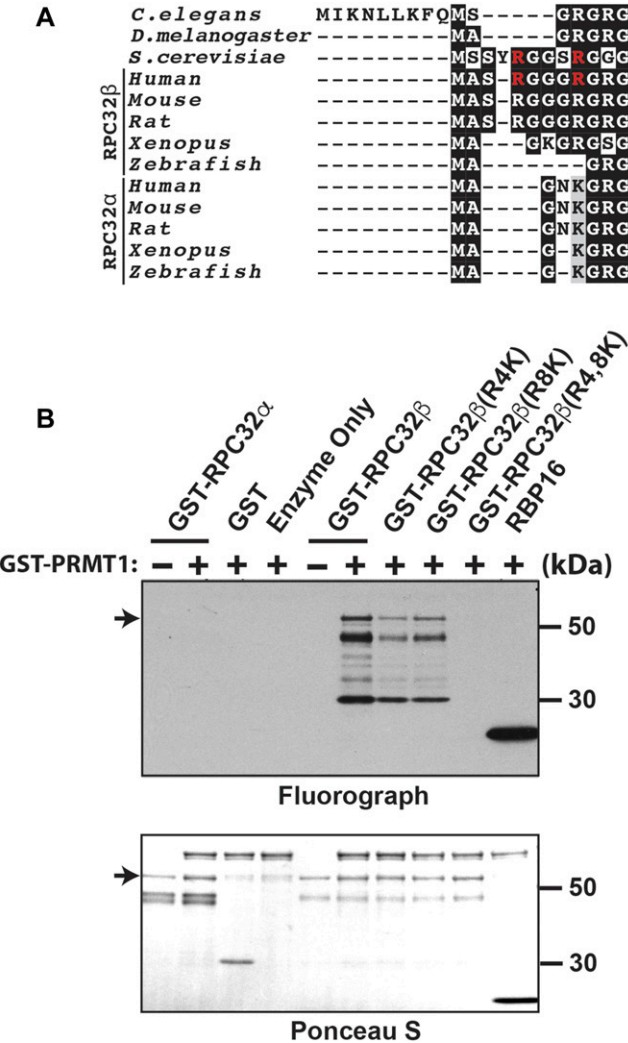

**Figure 7. Mammalian PRMT1 methylates human Rpc31 homolog RPC32β, but not RPC32α.**
**(A)** Sequence alignment of N-terminal amino acid sequences of yeast Rpc31 with its homologs in *Caenorhabditis elegans*, *Drosophila melanogaster*, and various mammalian species. Arginines on yeast Rpc31 that are colored red represent the identified methylated arginine and are conserved in RPC32β. **(B)** WT RPC32α and RPC32β were purified from *E. coli* and then subjected to in vitro methylation by rat PRMT1 and [methyl-³H]-SAM. In parallel, methylarginine substitution mutants of RPC32β (R4K, R8K and R4, 8K) were also tested. The arrow on the fluorograph denotes methylated RPC32β. RBP16 served as a positive control for the in vitro methylation. The protein loading levels for each sample are shown by Ponceau S staining of the same membrane before fluorography, with the arrow denoting the substrate tested.

the degree by which Rpc31's "stalk bridge" contacts RNA Pol III stalk and this, in turn, may impact the ability of RNA Pol III to carry out transcription at its maximum capacity.

### Biological significance stemming from a change in tRNA biogenesis

tRNAs are among the most abundant molecules in a cell. They are heavily modified by a large network of proteins that collectively process the pre-tRNA into a stable, mature tRNA (reviewed in Phizicky and Hopper (2010)). In addition, some tRNA modifications can be altered by changing Pol III activity, and this can lead to non-uniform changes in tRNA function that might have an impact on the translation profiles of specific mRNAs (Arimbasseri et al, 2016). Whereas a lack of Rpc31 methylation does not seem to be detrimental under conditions optimal for growth, it is possible that it alters the tRNA modifications and the translational profile that is required for an appropriate response to stress. On the other hand, an increase in abundance of a tRNA due to a defect in arginine methylation of Rpc31 could potentially lead to the futile cycling of tRNAs, thereby leading to wasted expenditure of energy as was previously observed in mice lacking the repressor MAF1 (Bonhoure et al, 2015).

### A working model for how RPC32β methylation controls cellular homeostasis

Under normal growth conditions, RPC32β is ubiquitously expressed but not RPC32α (Haurie et al, 2010). In undifferentiated and transformed cells, however, RPC32α expression is the highest (Haurie et al, 2010). The data from our yeast work show that Rpc31 methylation promotes its repression by Maf1 during stress. Therefore, the existence of two mammalian Rpc31 isoforms in which one can be methylated (RPC32β) but not the other (RPC32α) suggests a potential interplay between these isoforms in orchestrating a desired functional outcome in cellular homeostasis. One possible scenario for this interplay is that the level of methylated versus unmethylated RPC32 shifts the levels of Pol III isoforms available for MAF1-mediated repression. In other words, cells that express a high level of RPC32α, which lacks the methylation motif required for robust repression by MAF1, will dilute the overall level of RPC32β-incorporated Pol III isoforms available. As a consequence, there will be an increase in the level of RPC32α-incorporated Pol III isoforms in these cells. Isoforms of Pol III that incorporate either RPC32α or RPC32β target the same genes (Renaud et al, 2014), but having RPC32α-incorporated Pol III isoforms will make this isoform less susceptible to regulation by MAF1. In tumor cells, having more RPC32α-incorporated Pol III isoform will render these isoforms refractory to MAF1-mediated repression, thereby meeting the high demands of RNA Pol III transcription in tumor cells. Indeed, MAF1 expression and activity inversely correlates with the oncogenic activity (Shor et al, 2010; Palian et al, 2014). Given the key roles of RPC32α on cell differentiation and transformation, future experiments dissecting at how methylation contributes to RPC32 function in these processes may offer novel avenues of therapeutics targeting tumorigenesis.

## Materials and Methods

### Yeast strains used in this study

All yeast strains used are listed in Table S1. All plasmids used are listed in Table S2. All primers used are listed in Table S3. Cells were grown at 30°C on YPD medium except for the RNA hybridization assay of *hmt1Δ* cells depicted in Fig 2F, in which these cells were

grown at 30°C on SC + glucose. Genomic deletions were generated, and epitope tag cassettes were integrated, as previously described (Yu et al, 2004).

## Tandem-affinity purification of Rpc82

Tandem-affinity purification of TAP-tagged Rpc82 from *hmt1Δ* cells was carried out exactly as described previously (Jackson et al, 2012). The purified protein was dialyzed in 1× PBS/15% glycerol overnight, and the dialyzed fraction was then concentrated in a 10-kD MWCO Amicon Ultra 0.5 ml protein concentrator (Millipore). The concentrated samples were stored at −80°C before use.

## Rpc31-MORF overexpression

For MORF plasmid-containing yeast strains, the cells were first cultured SC minus uracil, and then the expression of MORF-tagged Rpc31 was induced for 6 h by the addition of 2× YEP with 2% sucrose and 2% galactose. To determine the relative levels of MORF-Rpc31 to endogenous Rpc31, diluted lysates made from induced yeast cultures were resolved by the SDS–PAGE, transferred to a nitrocellulose membrane, and then blotted with antibodies against Rpc31.

## In vitro methylation assay

Yeast Rpc31 was expressed using the yeast MORF collection (Open BioSystems) and purified as previously described (Gelperin et al, 2005). Methylarginine mutants of Rpc31 were expressed as GST-tagged fusion proteins and purified as previously described (Muddukrishna et al, 2017). The GST tag was cleaved off with thrombin and in vitro methylation assays were performed as previously described (Jackson & Yu, 2014).

## RNA hybridization assay

Yeast strains were grown to log phase (OD$_{600}$ ≈ 1.8) and one-half of the culture was treated with CPZ at a final concentration of 500 $\mu$M for 1 h. Both treated and untreated samples were harvested, and total RNA was extracted and RNA hybridization was carried out as previously described (Milliman et al, 2012).

## ChIP

Yeast strains were grown to log phase (OD$_{600}$ ≈ 1.8), and one-half of the culture was treated with CPZ at a final concentration of 500 $\mu$M for 1 h. ChIP was performed on both treated and untreated cells as previously described (Muddukrishna et al, 2017), with the exception of the sonication conditions (Branson Digital Sonifier 450, 3 mm tapered microtip, 20% amplitude, 20 s pulse/55 s pause/15× cycles). For each IP, anti-Myc (MS127P; Thermo Fisher Scientific) or anti-IgG (209-005-082; Jackson ImmunoResearch) antibodies were pre-coupled to protein A-Sepharose beads. qPCR was performed as previously described (Muddukrishna et al, 2017).

## RT–qPCR

RT–qPCR is carried out as previously described (Muddukrishna et al, 2017). Statistical testing was performed using ANOVA and post hoc Tukey's test, using the R statistical analysis software. All values reported are the mean of three biological replicates (n = 3).

## Yeast CoIP

Yeast strains were grown to log phase (OD$_{600}$ ≈ 1.8) and one-half of the culture was treated with CPZ for 1 h. The $\alpha$-Rpc31 antibody was cross-linked to magnetic beads (Dynabeads Cat. No. 14203) using the manufacturer's protocol. Lysates were generated as previously described (Muddukrishna et al, 2017). The lysates were brought to a total protein concentration of 10 mg/ml and incubated with $\alpha$-Rpc31–cross-linked beads for 2 h at 4°C, followed by three washes with PBS, 0.5% Triton X-100, and 2.5 mM MgCl$_2$. Bound protein was eluted in SDS gel loading buffer and resolved on a 4–12% Bis-Tris gradient gel (Life Technologies). The protein was transferred to nitrocellulose membrane, which was probed with relevant antibodies, developed using the Clarity Western ECL kit (Bio-Rad), imaged using the Bio-Rad ChemiDoc, and quantified using the Bio-Rad ImageQuant.

# Supplementary Information

# Acknowledgements

We thank Ian Willis for the Maf1-7SA and Maf1-Myc plasmids, and other reagents; Hung-Ta Chen for Rpc34 and Rpc160 antibodies; and Seok Kooi Khoi for helpful discussions. This work was supported by a National Science Foundation award (MCB-1051350) to MC Yu.

## Author Contributions

RB Davis: conceptualization, data curation, formal analysis, validation, investigation, methodology, and writing—original draft, review, and editing.
N Likhite: data curation, formal analysis, investigation, and methodology.
CA Jackson: data curation, formal analysis, validation, investigation, methodology, and writing—review and editing.
T Liu: formal analysis and methodology.
MC Yu: conceptualization, resources, data curation, formal analysis, supervision, funding acquisition, validation, investigation, methodology, project administration, and writing—original draft, review, and editing.

## Conflict of Interest Statement

The authors declare that they have no conflict of interest.

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
