## [Reviewer comments · Life Science Alliance]

Life Science Alliance

Robust Repression of tRNA Gene Transcription During Stress Requires Protein Arginine Methylation

Richoo Davis, Neah Likhite, Christopher Jackson, Tao Liu, and Michael Yu

DOI: <https://doi.org/10.26508/lsa.201900261>

Corresponding author(s): Michael Yu, SUNY-Buffalo

Review Timeline:	Submission Date:	2018-11-29
	Editorial Decision:	2018-12-19
	Revision Received:	2019-05-02
	Editorial Decision:	2019-05-16
	Revision Received:	2019-05-21
	Accepted:	2019-05-22

Scientific Editor: Andrea Leibfried

Transaction Report:

December 19, 2018

Re: Life Science Alliance manuscript #LSA-2018-00261-T

Michael C Yu
SUNY-Buffalo

Dear Dr. Yu,

Thank you for submitting your manuscript entitled "Robust Repression of tRNA Gene Transcription During Stress Requires Protein Arginine Methylation" to Life Science Alliance. The manuscript was assessed by expert reviewers, whose comments are appended to this letter.

As you will see, the reviewers note that there is value in your findings, while the insight into a controlled regulation of Rpc31 methylation and thus transcriptional output regulation is missing. However, all three reviewers note that the chosen approach to data normalization differs from your previous work and that the discrepancy observed remains unexplained.

Given the reviewers' input, we would like to invite you to submit a revised version of your manuscript, addressing all technical issues noted, especially the crucial issues of data normalization to make a rational explanation for the reported effects of hmt1 deletion/Rpc31 mutation with respect to the previously reported results. Furthermore, the requested controls should get added, and further validation of the conclusions should get provided. Please note that we will need strong support from the reviewers on such a revised version. Data on the / a physiological significance of your findings, however, do not need to get added in the revision.

Thank you for this interesting contribution to Life Science Alliance. We are looking forward to

receiving your revised manuscript.

Sincerely,

- A letter addressing the reviewers' comments point by point.
- An editable version of the final text (.DOC or .DOCX) is needed for copyediting (no PDFs).
- High-resolution figure, supplementary figure and video files uploaded as individual files: See our detailed guidelines for preparing your production-ready images, <http://life-science-alliance.org/authorguide>
- Summary blurb (enter in submission system): A short text summarizing in a single sentence the study (max. 200 characters including spaces). This text is used in conjunction with the titles of papers, hence should be informative and complementary to the title and running title. It should describe the context and significance of the findings for a general readership; it should be written in the present tense and refer to the work in the third person. Author names should not be mentioned.

B. MANUSCRIPT ORGANIZATION AND FORMATTING:

Full guidelines are available on our Instructions for Authors page, <http://life-science-alliance.org/authorguide>

Reviewer #1 (Comments to the Authors (Required)):

Rpc31 is an essential subunit of Pol III complex with no counterpart in the Pol II or Pol I transcription systems. The submitted manuscript from the laboratory of Michael C. Yu describes the discovery of Rpc31 methylation by Hmt1 methyltransferase and addresses its effect on transcription of tRNA genes. Pol III regulation by posttranslational modifications of its subunits is an area of importance and general interest, appropriate for Life Science Alliance. The authors have made strong contributions to the field, since a role of the modification of Rpc31 subunit has not been reported so far. Moreover, while recent studies in the field highlighted the regulatory role of phosphorylation and sumoylation of Pol III subunits, the function of methylation has not been explored.

The association of Hmt1 methyltransferase with Pol III chromatin has been shown previously in the laboratory of Michael C. Yu and published by Milliman and colleagues (BMC Genomics, 2012). The paper reported also increased tRNA levels in Hmt1 loss-of-function mutants and the Hmt1 interaction with the TFIIIB component, Bdp1, indicating a regulatory role of Hmt1 in Pol III transcription.

The current manuscript provides several novel aspects of Pol III regulation: 1. Identification of Rpc31 subunit of Pol III as a target of Hmt1 methyltransferase; 2. Demonstration, that the inactivation of the methylation sites in Rpc31 protein decreases the interaction between Pol III and Maf1, and reduces Maf1 association with Pol III genes; 3. Demonstration that the human ortholog of yeast Rpc31, RPC32b, is methylated by PRMT1 in vitro.

In my opinion, however, the proposed function of Rpc31 methylation in the adaptation of Pol III transcription levels to growth conditions requires deeper experimental validation. A few conclusions are not supported by the provided data (detailed comments are provided below). Presenting some original data and performing additional experiments would certainly provide an additional insight that would warrant the publication in Life Science Alliance.

Major comments:

1. According to the authors, Hmt1 is required to promote transcription of tRNA genes under optimal growth conditions (Fig. 2 D-E) and to repress transcription of tRNA genes under stress (Fig. 3A). This unusual regulatory role of Hmt1 in tRNA transcription is insufficiently supported by the experimental data. Both mentioned figures show only quantification of hybridization experiments. Additionally, the results presented in the previous paper (BMC Genomics, 2012, Fig. 3C) are inconsistent with these shown here (Fig. 2C) and the authors explained this discrepancy by the differences in the applied loading controls. Presentation of the original hybridization blots with the appropriate loading controls is absolutely required. RNA isolated from wt and hmt1 Δ strains, grown in optimal and stress conditions, should be analyzed by Northern blotting with the probes, which detect primary tRNA transcripts, intron-containing tRNA precursors or/and other probes which also recognize mature tRNA. Amounts of primary transcripts should be normalized to the loading control and calculated relative to the amounts in the wt strain under standard conditions.

2. Fig. 1 presents data of ChIP experiment in the correct form; the occupancy is represented as a percentage of immunoprecipitation over input chromatin. Rpc82 and Rpc160 occupancy (Fig. 3B) and Maf1 occupancy (Fig. 4D) of candidate tRNA genes in wt strain and mutants is illustrated only by the Post hoc Tukey's Honest Significant Differences method. Presentation of this data as a percentage of immunoprecipitation over input chromatin, including statistical analysis, is absolutely required.

3. Description of Fig. 1 "Hmt1 occupancy at tRNA genes is dependent on the transcriptional activity of the gene" is not appropriate because only decrease of Hmt1 occupancy, but not decrease of transcription under stress conditions is presented. Moreover, panels A, B, C show decrease of Hmt1 occupancy in cells treated with various stress but untreated control samples markedly differ between experiments. E.g. the differences in Hmt1 occupancy at tR(UCU)D and tY(GUA)D between panel B and C are over 100%. The authors should explain the differences of Hmt1 occupancy on

the same tRNA genes within control samples.

4. In the previous paper (BMC Genomics, 2012) the authors showed, using coimmunoprecipitation assay, that Hmt1 interacts with Bdp1 subunit of TFIIIB. Does Hmt1 interact physically with Rpc31 subunit of Pol III complex as well? It should be experimentally addressed. Alternatively, possible effect of Bdp1-Hmt1 interaction on Rpc31 methylation should be discussed on the basis of the known structure of Pol III initiation complex.

Reviewer #2 (Comments to the Authors (Required)):

This work explores the role of arginine methylation in transcription by RNA polymerase (Pol) III as follows a genome-wide occupancy study from the same group showing that Pol III loci are major targets of the arginine methyltransferase Hmt1 in yeast. The current work identifies the Rpc31 subunit of Pol III as a substrate of Hmt1 in vitro. The experiments also examine the consequences of deleting HMT1 and of an arginine methylation defective Rpc31 mutant for Pol III occupancy and transcription in normal and stressed cells. Curiously, the authors find that arginine methylation of Rpc31 contributes positively to transcription under optimal growth conditions and negatively to transcription under repressing conditions. A major concern is that the later interpretation is critically dependent on how the data are processed and since the details are lacking, it is not possible to properly evaluate the conclusions. Even if the conclusions are borne out, the role of Hmt1 in regulating Pol III transcription remains unclear since the system does not appear to be dynamically controlled by Hmt1 (contrary to the subheading in the article), i.e. no differential in arginine methylation of Rpc31 has been demonstrated under different conditions. This in turn raises questions about the significance of the modest effect of arginine methylation on Pol III binding to Maf1, as demonstrated by IP. This negative view needs to be weighed against the finding, also reported in this study, that arginine methylation of the Rpc31 ortholog in humans is conserved, at least for one of the isoforms (RPC32 β). The suggestion that human Pol III containing the RPC32 α isoform, which predominates in undifferentiated and transformed cells, might be more resistant to repression by Maf1 is interesting and potentially important.

Important issues

Fig. 2B The ability of recombinant Hmt1 to methylate Rpc31 in vitro is clearly demonstrated. However, the conditions of this experiment involve overexpression of Rpc31 from a MORF library plasmid (i.e. high copy number and Gal promoter). Under these conditions most of the Rpc31 is presumably not present as part of the Pol III complex. This is difficult to ascertain since no blots were presented to show the amount of Rpc31 in overexpressed versus non-overexpressed cells. The need to overexpress Rpc31 to demonstrate arginine methylation raises a question of physiological relevance that the authors should address.

Fig 2 D and E. In unstressed cells, pre-tRNA levels normalized to U4 snRNA go down in hmt1 Δ and Rpc31 R5,9A mutant strains. Since these results are opposite to those reported previously when 5.8S RNA was used for normalization, it seems prudent to employ an independent measure to ensure the result is now correct. A 3H-uracil pulse chase experiment is probably the best option but normalization of blots to another internal standard would probably suffice. The earlier experiments indicated that mature tRNA levels were elevated in the hmt1 Δ strain. Is this result reversed as well when using U4 for normalization? What is the status of mature tRNAs in the hmt1 Δ strain?

Fig 3. The representation of the data is not clear. "Fold decrease over untreated", plotted on the x-

axis, is confusing. What is the quantitative effect represented by a one-fold change? Since repression is being plotted in panel A, it would make more sense to indicate the reduction relative to the untreated wild-type with the latter defined as 100% for the different genes. Also not clear is whether the values for the mutants are relative to the untreated mutant (which would be misleading in this case because the untreated mutants are already lower than 100%) or relative to the untreated WT. This needs to be specified. How were the distributions determined, what does the height of the distribution reflect and is this an appropriate representation of the data? Similar normalization issues apply to the Chip data in panel B.

Related to this issue, what does it mean for repression to be lower in the mutants? Diminished repression is easily achieved and incorrectly interpreted when untreated mutants that have lower transcription than WT are referenced at 100%. Details on how the data are processed are needed to resolve this concern. This information is critical in evaluating the conclusions that have been drawn.

Minor issues

P 3. "Pol III is recruited to tRNA genes by two general transcription factors, TFIIIB and TFIIIC. The latter is a complex that recognizes sequence-specific promoter elements and guides the concerted binding of three TFIIIB subunits (TBP, Brf1, and Bdp1) to these elements." Although the text does not mention specific promoter elements, the subject is clearly the A and B blocks since these are the elements recognized by TFIIIC. The authors are well aware that "these elements" are not also bound by the TFIIIB subunits as implied by the context, yet the sentence, as written, conveys this incorrect meaning.

P 5. "Based on these observations, we conclude that the association of Hmt1 with tRNA genes is dependent on the levels of transcription of the latter." Couldn't high levels of transcription be dependent on Hmt1? It is not clear whether Hmt1 associates with Pol III genes because they are transcriptionally active (i.e. a dependent response) or whether the association of Hmt1 contributes to the activity of the genes (i.e. a causal effect). What is clear is that the association of Hmt1 is correlated with gene activity. Perhaps it should be stated in this way.

P 7. Making a Pol II transcript a potentially better....

P 8. Investigating repression of tRNA biogenesis by stress in the *hmt1Δ* and *Rpc31* mutant strains.

P 9. we compared the distribution of the fold decreases in Pol III occupancy after treatment with CPZ and found that the repression of Pol III occupancy was significantly lower....." Need to clarify/reexpress.

P 12. Our yeast data show that a lack of arginine methylation of *Rpc31* can result in diminished repression of Pol III activity.

P 12. The roles of arginine methylation of *Rpc31* in controlling Pol III transcription is dynamic - no condition-dependent methylation dynamics have been demonstrated.

P 12. "However, one can argue that if arginine methylation of a substrate can promote distinct outcomes for a biological process (e.g. Pol III transcription) based on the specific trigger (e.g. nutrient or stress signaling), a lack-of-turnover mechanism would result in higher efficiency and specificity in tuning that process." Are there any examples that support this logic?

P 13. Thus, it is possible that Rpc31 methylation plays both positive and negative roles in Pol III transcription by participating in distinct biochemical interactions.

Reviewer #3 (Comments to the Authors (Required)):

In this manuscript, Davis et al. follow up on their previous publication that described the association of the protein arginine methyltransferase hmt1 in with the majority of tRNA genes in the yeast *S. cerevisiae*. In the paper published by Milliman et al. in 2012, they also described that the loss of hmt1 activity was associated with higher tRNA expression levels.

Here, they have undertaken an effort to mechanistically explain the observations published by Milliman et al. in 2012. It was described by others that tRNAs are transcribed by RNA polymerase III (Pol III) and that subunits RPC53 and RPC31 of Pol III are methylated. Davis et al. identified RGG tripeptides and RG repeats in RPC31 and showed that RPC31 was methylated in vivo in an hmt1-dependent manner. They identified arginines 5 and 9 of RPC31 as the targets of hmt1. By changing the method of normalization of gene expression, they found that loss of hmt1 activity lead to decreased tRNA levels rather than enhanced as they described in their earlier paper. The same observation of reduced tRNA levels was made with a strain expressing the R5,9A mutant of RPC31. Pol III transcription is reduced under conditions of stress such as nutrition starvation, chlorpromazin (CPZ) or tunicamycin treatment. Davis et al. showed that loss of hmt1 activity or RPC31 R5,9A mutation reduced the levels of Pol III transcription repression under CPZ stress. This effect was associated with reduced interaction of Pol III and Maf1.

Finally, Davis et al. analyzed whether one of the mammalian homologues of RPC31 could be methylated by PRMT1, the rat orthologue of hmt1. They showed that RPC32 β , but not RPC32 α contained methylated arginines at positions 4 and 8.

This manuscript describes a novel type of modification of an RNA polymerase III subunit and proposes functional effects of this modification on Pol III transcription. The finding in itself is interesting, but several questions remain unanswered with regard to the regulation of tRNA expression by RPC31 methylation.

Major concerns :

The data of the authors suggest that methylation of RPC31 has an anti-repressive function on Pol III transcription, possibly mediated by loss of interaction with Maf1, at least under unfavorable growth conditions. However, it remains entirely unclear how methylation of RPC31 is regulated and what its function is in regulating Pol III transcription. The authors mention that no demethylase has been identified, indicating that this modification represents a point without return. Therefore, it is hard to imagine that it serves for adapting Pol III transcription levels to distinct growth conditions. In this context, the anti-repressive function of RPC31 methylation described here does not really make any sense.

The authors changed their method of normalizing tRNA expression. Instead of the 5.8 S rRNA, they utilize U4 snRNA. All results are dependent on this choice and are in conflict with their own previously published data. What is the rationale other than citing another publication for being convinced that this normalization is more valid than the one employed in the Milliman paper? Is U4 expression stable under their conditions and if so, how has this been determined? I believe that higher standards of normalization should be employed. Two or more genes should be included into

the analysis for obtaining more reliable data on the expression of tRNAs. Several different algorithms have been proposed for evaluating gene expression and determining best reference genes (for instance geNorm, NormFinder or BestKeeper).

Except for the data shown in Figure 1, CPZ treatment is presented in this manuscript as equivalent to cell stress. It may elicit unique effects that are not observed upon other forms of stress. Therefore, it would be important not only to determine whether nutrient starvation or tunicamycin treatment generate the same effects on Pol III transcription (shown in Fig 1), but also to analyze whether loss of hmt1 activity or R5,9A mutation of RPC31 result in lower repression of tRNA expression under these forms of stress?

Hmt1 also interacts with the B' subunit of TFIIB. Does this interaction have an influence on the observed effects on tRNA expression?

Minor concerns :

The authors show in Figure 1 that the transcription of the 5S gene is not altered upon starvation, CPZ or tunicamycin treatment. Is the same true for deletion of hmt1 activity or RPC31 R5,9A mutation? Is the Pol III transcribed 5S gene differently affected by RPC31 methylation compared to tRNA genes?

The finding that RPC32 β , but not RPC32 α is modified by methylation is counterintuitive with respect to their supposed functions in cellular homeostasis and stem cell- or tumor-supporting activities. RPC31 methylation is associated with increased tRNA levels during normal or optimal growth and with protection from repression under cellular stress. Therefore, it does not make sense that the homeostasis-regulating subunit RPC32 β is associated with this tRNA level enhancing modification. It would make sense if the tumor-associated RPC32 α subunit would be methylated, thereby protecting tumor cells from reduced tRNA levels during stress and allowing high expression levels under normal growth conditions, thereby favoring tumor growth. However, the data on methylation of RPC31 and RPC32 β seem to be solid and I believe that the functional outcome of this modification needs to be better characterized for publication of these data. See also first comment in major concerns.

The Wang & Roeder paper published in *Genes & Development* in 1997 is cited for the description of the yeast RPC31-RPC34-RPC82 complex. However, this paper describes the human RPC32-RPC39-RPC62 complex. A more appropriate citation would be Werner et al., 1992, also supported by the data published by Thuillier et al., 1995.

Response to Reviewer #1

Major comments:

1. According to the authors, Hmt1 is required to promote transcription of tRNA genes under optimal growth conditions (Fig. 2 D-E) and to repress transcription of tRNA genes under stress (Fig. 3A). This unusual regulatory role of Hmt1 in tRNA transcription is insufficiently supported by the experimental data. Both mentioned figures show only quantification of hybridization experiments. Additionally, the results presented in the previous paper (BMC Genomics, 2012, Fig. 3C) are inconsistent with these shown here (Fig. 2C) and the authors explained this discrepancy by the differences in the applied loading controls. Presentation of the original hybridization blots with the appropriate loading controls is absolutely required. RNA isolated from wt and *hmt1Δ* strains, grown in optimal and stress conditions, should be analyzed by Northern blotting with the probes, which detect primary tRNA transcripts, intron-containing tRNA precursors or/and other probes which also recognize mature tRNA. Amounts of primary transcripts should be normalized to the loading control and calculated relative to the amounts in the wt strain under standard conditions.

Author's Response: To address the concerns raised by the reviewer, we have repeated this experiment and normalized our data using two additional loading controls: U5 and U3. Both U5 and U3 have been previously used by other labs as loading controls in RNA hybridization experiments. In our RNA hybridization experiments normalized with either U5 or U3, we obtained data that are similar in trend as those with U4 in terms of wild-type vs either *hmt1Δ* or Rpc31 mutant. This data is now presented as Figs. 3A and 3B in the revised manuscript.

For the *hmt1Δ* data depicted in the original Figure 2D, the RNA used was extracted from cells grown in synthetic complete (SC) + glucose. In our original work, we carried out this experiment in both YPD and SC+glucose but the differences seen in cells grown in YPD were smaller or not as consistent when compared to SC+glu grown cells. When *hmt1Δ* cells were grown in SC+glu, we consistently see a decrease in the levels of pre-tRNAs to be lower in the *hmt1Δ* cells across all four precursor tRNAs tested when compared to the wild-type cells. For Rpc31 mutant, the differences were apparent and consistent in YPD grown cells. Since all the subsequent experiments in this paper were done with YPD-grown cells, we have repeated our RNA hybridization experiment using RNA extracted from cells grown in YPD. We have presented this additional data and rationale in our revised manuscript. As per Reviewer #2's suggestion, this new figure compares the level of reduction relative to the untreated wild-type (which is defined as 100% for the different genes). In addition, we have presented samples of the original hybridization blots, as well as signals from all the loading controls used (U4, U5, and U3). This is now presented as Supplemental Fig. 1.

The tRNAs we analyzed were chosen because these tRNAs were bound by Hmt1 and they also allow us to use probes that would specifically detect only the primary tRNA transcripts (or partially processed). Given our focus is on the transcription of precursor

tRNAs, we do not feel that additional information gathered from the mature tRNAs would strengthen our case as there are extensive processing events that occur between primary to mature tRNA. Thus, any perceived differences would require further experimental clarification to rule this out. While the reviewer's point on the mature tRNAs is interesting and important, we feel these experiments would fall outside the scope of our current manuscript.

2. Fig. 1 presents data of ChIP experiment in the correct form; the occupancy is represented as a percentage of immunoprecipitation over input chromatin. Rpc82 and Rpc160 occupancy (Fig. 3B) and Maf1 occupancy (Fig. 4D) of candidate tRNA genes in wt strain and mutants is illustrated only by the Post hoc Tukey's Honest Significant Differences method. Presentation of this data as a percentage of immunoprecipitation over input chromatin, including statistical analysis, is absolutely required.

Author's Response: We have changed the way we depict the ChIP data to present it as percentage input as per reviewer's suggestion. For statistical significance, we performed Student's *t*-test to compare the protein occupancy between strains and between treated vs. untreated samples in each strain. The use of Tukey's HSD in determining the significance of the changes seen among the three strains is based on the recommendation from our biostatistician, this statistical test provides the required statistical support to lay the claim on the differences in CPZ response between strains. To better help readers better understand our rationale and interpretation, we have revised our explanation in the text for how the Tukey's HSD test helps us determining the significance of changes seen across all three strains (wild-type, *hmt1Δ*, and Rpc31 mutant).

3. Description of Fig. 1 "Hmt1 occupancy at tRNA genes is dependent on the transcriptional activity of the gene" is not appropriate because only decrease of Hmt1 occupancy, but not decrease of transcription under stress conditions is presented.

Author's Response: We have revised our description in Fig. 1 to "Hmt1 occupancy at tRNA genes is decreased under stress conditions".

Moreover, panels A, B, C show decrease of Hmt1 occupancy in cells treated with various stress but untreated control samples markedly differ between experiments. E.g. the differences in Hmt1 occupancy at tR(UCU)D and tY(GUA)D between panel B and C are over 100%. The authors should explain the differences of Hmt1 occupancy on the same tRNA genes within control samples.

Author's Response: We agree with the reviewer that there is a discrepancy with the level of Hmt1 occupancy between Fig. 1 panels in the untreated cells. We believe this is due to technical issues that stem from variations in IPs from each set of experiments, as we use % input on the figure legend rather than normalizing against a "control gene". This is supported by our 5S control and No ORF control values which were also higher,

suggesting an overall higher background signal that may account for a higher Hmt1 occupancy signal in such experiment. Nevertheless, the overall trend for a decrease in Hmt1 occupancy in cells treated with various stressors remains the same. We have added this explanation in our revised manuscript.

4. In the previous paper (BMC Genomics, 2012) the authors showed, using coimmunoprecipitation assay, that Hmt1 interacts with Bdp1 subunit of TFIIIB. Does Hmt1 interact physically with Rpc31 subunit of Pol III complex as well? It should be experimentally addressed. Alternatively, possible effect of Bdp1-Hmt1 interaction on Rpc31 methylation should be discussed on the basis of the known structure of Pol III initiation complex.

Author's Response: We have performed a co-immunoprecipitation assay to address the reviewer's concern. Using our antibody against Rpc31, we show that Hmt1 is able to be co-immunoprecipitated. This data is now presented as Fig. 5D in our revised manuscript.

Our hypothesis for the role of Bdp1's interaction with Hmt1 is that Bdp1 recruits Hmt1 to the Pol III transcription system in order to promote Hmt1's ability to methylate RNA Pol III subunit Rpc31. The N-terminus of Rpc31 is a disordered region that is solvent exposed and likely to be within the range for Hmt1-mediated methylation.

Response to Reviewer #2:

Important issues

Fig. 2B The ability of recombinant Hmt1 to methylate Rpc31 in vitro is clearly demonstrated. However, the conditions of this experiment involve overexpression of Rpc31 from a MORF library plasmid (i.e. high copy number and Gal promoter). Under these conditions most of the Rpc31 is presumably not present as part of the Pol III complex. This is difficult to ascertain since no blots were presented to show the amount of Rpc31 in overexpressed versus non-overexpressed cells. The need to overexpress Rpc31 to demonstrate arginine methylation raises a question of physiological relevance that the authors should address.

Author's Response: We used the MORF system as a means to purify Rpc31 because the C-terminal tagging of Rpc31 was detrimental to the growth of cells. As for N-terminal tagging, we were concerned that such tagging may affect the capability of Rpc31 to be methylated since the N-terminus is where the methylation motif is located. To address the reviewer's concern, we carried out an immunoblot analysis to determine the amounts of MORF-Rpc31 relative to the endogenous levels of Rpc31. This data is now presented as Fig. 2B in the revised manuscript. Interestingly, the MORF-Rpc31 purified from the *HMT1* cells can no longer undergo further methylation unlike the MORF-Rpc31 purified from *hmt1Δ* cells, which are presumed to be in a hypomethylated form (Fig. 2C). This observation suggests that, despite a much higher levels of MORF-Rpc31 present over the endogenous Rpc31, Hmt1 is still able to methylate these MORF-Rpc31 effectively.

To address the physiological relevance of Rpc31 methylation, we purified TAP-tagged Rpc82 from *hmt1Δ* cells and subjected co-purified proteins, which includes Rpc31, to *in vitro* methylation. Our data show that Hmt1 can methylate Rpc31 when it is presented in a complex with Rpc82 and Rpc34, which resembles a more physiologically relevant condition. This data is now presented as Fig. 2D in the revised manuscript.

Fig 2D and E. In unstressed cells, pre-tRNA levels normalized to U4 snRNA go down in *hmt1Δ* and *Rpc31 R5,9A* mutant strains. Since these results are opposite to those reported previously when 5.8S RNA was used for normalization, it seems prudent to employ an independent measure to ensure the result is now correct. A 3H-uracil pulse chase experiment is probably the best option but normalization of blots to another internal standard would probably suffice. The earlier experiments indicated that mature tRNA levels were elevated in the *hmt1Δ* strain. Is this result reversed as well when using U4 for normalization? What is the status of mature tRNAs in the *hmt1Δ* strain?

Author's Response: To address the concerns raised by the reviewer, we have repeated this experiment and normalized our data using two additional loading controls: U5 and U3. Both U5 and U3 have been previously used by other labs as loading controls in

RNA hybridization experiments. In our RNA hybridization experiments normalized with either U5 or U3, we obtained data that are similar in trend as those with U4 in terms of wild-type vs either *hmt1Δ* or Rpc31 mutant. This data is now presented as Figs. 3A and 3B in the revised manuscript.

For the *hmt1Δ* data depicted in the original Figure 2D, the RNA used was extracted from cells grown in synthetic complete (SC) + glucose. In our original work, we carried out this experiment in both YPD and SC+glucose but the differences seen in cells grown in YPD were smaller or not as consistent when compared to SC+glu grown cells. When *hmt1Δ* cells were grown in SC+glu, we consistently see a decrease in the levels of pre-tRNAs to be lower in the *hmt1Δ* cells across all four precursor tRNAs tested when compared to the wild-type cells. For Rpc31 mutant, the differences were apparent and consistent in YPD grown cells. Since all the subsequent experiments in this paper were done with YPD-grown cells, we have repeated our RNA hybridization experiment using RNA extracted from cells grown in YPD. We have presented this additional data and rationale in our revised manuscript. As per reviewer's suggestion, this new figure compares the level of reduction relative to the untreated wild-type (which is defined as 100% for the different genes). In addition, we have presented samples of the original hybridization blots, as well as signals from all the loading controls used (U4, U5, and U3). This is now presented as Supplemental Fig. 1.

The tRNAs we analyzed were chosen because these tRNAs were bound by Hmt1 and they also allow us to use probes that would specifically detect only the primary tRNA transcripts (or partially processed). Given our focus is on the transcription of precursor tRNAs, we do not feel that additional information gathered from the mature tRNAs would strengthen our case as there are extensive processing events that occur between primary to mature tRNA. Thus, any perceived differences would require further experimental clarification to rule this out. While the reviewer's point on the mature tRNAs is interesting and important, we feel these experiments would fall outside the scope of our current manuscript.

Fig 3. The representation of the data is not clear. "Fold decrease over untreated", plotted on the x-axis, is confusing. What is the quantitative effect represented by a one-fold change? Since repression is being plotted in panel A, it would make more sense to indicate the reduction relative to the untreated wild-type with the latter defined as 100% for the different genes. Also not clear is whether the values for the mutants are relative to the untreated mutant (which would be misleading in this case because the untreated mutants are already lower than 100%) or relative to the untreated WT. This needs to be specified. How were the distributions determined, what does the height of the distribution reflect and is this an appropriate representation of the data? Similar normalization issues apply to the Chip data in panel B.

Related to this issue, what does it mean for repression to be lower in the mutants? Diminished repression is easily achieved and incorrectly interpreted when untreated mutants that have lower transcription than WT are referenced at

100%. Details on how the data are processed are needed to resolve this concern. This information is critical in evaluating the conclusions that have been drawn.

Author's Response: We have taken the reviewer's suggestion and re-plotted the graph for the RNA hybridization (for all three normalizing controls) to indicate the reduction relative to the untreated wild-type with the latter defined as 100% for the different genes.

The use of Tukey's HSD in determining the significance of the changes seen among the three strains is based on the recommendation from our biostatistician, this statistical test provides the required statistical support to lay the claim on the differences in CPZ response between strains. To better help readers better understand our rationale and interpretation, we have revised our explanation in the text for how the Tukey's HSD test helps us determining the significance of changes seen across all three strains (wild-type, *hmt1Δ*, and Rpc31 mutant).

In terms of the meaning behind the lower repression in the mutants, we feel it is more meaningful to interpret the data in a collective manner. When considering the changes seen in the RNA hybridization results (increased levels of pre-tRNAs for the mutants under stress vs. wild-type under stress), it is also important to consider the changes seen with RNA Pol III occupancy, Maf1 occupancy, and the physical association between Maf1 and RNA Pol III. When these data are considered together, we believe they collectively do support the conclusions drawn.

Minor issues

P 3. "Pol III is recruited to tRNA genes by two general transcription factors, TFIIB and TFIIIC. The latter is a complex that recognizes sequence-specific promoter elements and guides the concerted binding of three TFIIB subunits (TBP, Brf1, and Bdp1) to these elements." Although the text does not mention specific promoter elements, the subject is clearly the A and B blocks since these are the elements recognized by TFIIIC. The authors are well aware that "these elements" are not also bound by the TFIIB subunits as implied by the context, yet the sentence, as written, conveys this incorrect meaning.

Author's Response: we have changed our sentence to "Pol III is recruited to tRNA genes by two general transcription factors, TFIIB and TFIIIC. The latter is a complex that recognizes sequence-specific promoter elements and guides the concerted binding of three TFIIB subunits (TBP, Brf1, and Bdp1) to the transcription start site."

P 5. "Based on these observations, we conclude that the association of Hmt1 with tRNA genes is dependent on the levels of transcription of the latter." Couldn't high levels of transcription be dependent on Hmt1? It is not clear whether Hmt1 associates with Pol III genes because they are transcriptionally active (i.e. a dependent response) or whether the association of Hmt1 contributes to the activity of the genes (i.e. a causal effect). What is clear is that the association of Hmt1 is correlated with gene activity. Perhaps it should be stated in this way.

Author's Response: we have changed our sentence to "Based on these observations, we conclude that the association of Hmt1 with tRNA genes correlates to the levels of transcription of the latter."

P 7. Making a Pol II transcript a potentially better....

Author's response: we have changed our sentence to "making a Pol II transcript a potentially better means of assessing the impact of Hmt1 on the transcriptional activities of Pol III."

P 8. Investigating repression of tRNA biogenesis by stress in the *hmt1*Δ and *Rpc31* mutant strains.

Author's response: we have changed the heading of such results section to "Investigating the repression of tRNA biogenesis under stress in the *hmt1*Δ and *Rpc31*^{R5,9A} strains."

P 9. we compared the distribution of the fold decreases in Pol III occupancy after treatment with CPZ and found that the repression of Pol III occupancy was significantly lower....." Need to clarify/reexpress.

Author's response: to clarify this further, we have now expressed the ChIP data in terms of fold change over wild-type to display the levels of Pol III occupancy before and after treatment with CPZ. Furthermore, we have re-phrased our explanation for the violin plot to better describe the purpose of our analysis and the interpretation of our findings.

P 12. Our yeast data show that a lack of arginine methylation of *Rpc31* can result in diminished repression of Pol III activity.

Author's response: we have changed our sentence as per reviewer's suggestion.

P 12. The roles of arginine methylation of *Rpc31* in controlling Pol III transcription is dynamic - no condition-dependent methylation dynamics have been demonstrated.

Author's response: our data suggest such distinct possibility may exist in yeast and this mechanism would provide a rational approach for yeast to maximize the ways that would require functions carried out by a methylated *Rpc31*, without having to cycle through erasing and re-establishing the methyl mark.

P 12. "However, one can argue that if arginine methylation of a substrate can promote distinct outcomes for a biological process (e.g. Pol III transcription) based on the specific trigger (e.g. nutrient or stress signaling), a lack-of-turnover mechanism would result in higher efficiency and specificity in tuning that process." Are there any examples that support this logic?

Author's response: Most of the PTMs studied to date involves dynamic regulation of such PTM by enzymes. This is supported by well documented example such as kinase/phosphatase for phosphorylation or acetylase/deacetylase for acetylation. Arginine methylation is unique in a sense that no *bona fide* demethylase that can effectively reverse the methyl mark has ever been identified. In the case of Rpc31, methylation is both evolutionarily conserved and expensive from a metabolic point of view. Hence, we posit that by bypassing the need of a demethylase, the organism would actually achieve a higher specificity (since it is the same, rather than different, methyl marks, that can render different functional outcomes in a biological process) and efficiency (since the cells would bypass a need for a demethylase enzyme that would, in turn, catalyze the demethylation reaction to cycle through a methylation/demethylation process) in maximizing the use of arginine methylation to tune a biological process, where such process is susceptible to changes in the environment.

P 13. Thus, it is possible that Rpc31 methylation plays both positive and negative roles in Pol III transcription by participating in distinct biochemical interactions.

Author's response: we have changed our sentence as per reviewer's suggestion.

Response to Reviewer #3:

Major concerns:

The data of the authors suggest that methylation of RPC31 has an anti-repressive function on Pol III transcription, possibly mediated by loss of interaction with Maf1, at least under unfavorable growth conditions. However, it remains entirely unclear how methylation of RPC31 is regulated and what its function is in regulating Pol III transcription. The authors mention that no demethylase has been identified, indicating that this modification represents a point without return. Therefore, it is hard to imagine that it serves for adapting Pol III transcription levels to distinct growth conditions. In this context, the anti-repressive function of RPC31 methylation described here does not really make any sense.

Author's Response: One of the key challenges in the field of protein arginine methylation is the dynamics of the methylation mark. In yeast, no specific protein has been identified to have an arginine demethylase activity. While one interpretation is that this modification may represent a "point without return", another possibility is that an alternative mechanism may exist that utilize this modification in a different manner. This is because methylation comes at a fairly expensive metabolic cost to the cell and by simply turning the protein over would be wasteful to a cell's metabolic economy. What our data suggest is an alternative option to utilize a methylated protein in various contexts without cycling through a methylation/demethylation process.

The authors changed their method of normalizing tRNA expression. Instead of the 5.8 S rRNA, they utilize U4 snRNA. All results are dependent on this choice and are in conflict with their own previously published data. What is the rational other than citing another publication for being convinced that this normalization is more valid than the one employed in the Milliman paper? Is U4 expression stable under their conditions and if so, how has this been determined? I believe that higher standards of normalization should be employed. Two or more genes should be included into the analysis for obtaining more reliable data on the expression of tRNAs. Several different algorithms have been proposed for evaluating gene expression and determining best reference genes (for instance geNorm, NormFinder or BestKeeper).

Author's Response: To address the concerns raised by the reviewer, we have repeated this experiment and normalized our data using two additional loading controls: U5 and U3. Both U5 and U3 have been previously used by other labs as loading controls in RNA hybridization experiments. In our RNA hybridization experiments normalized with either U5 or U3, we obtained data that are similar in trend as those with U4 in terms of wild-type vs either *hmt1Δ* or *Rpc31* mutant. This data is now presented as Figs. 3A and 3B in the revised manuscript.

For the *hmt1Δ* data depicted in the original Figure 2D, the RNA used was extracted from cells grown in synthetic complete (SC) + glucose. In our original work, we carried out

this experiment in both YPD and SC+glucose but the differences seen in cells grown in YPD were smaller or not as consistent when compared to SC+glu grown cells. When *hmt1Δ* cells were grown in SC+glu, we consistently see a decrease in the levels of pre-tRNAs to be lower in the *hmt1Δ* cells across all four precursor tRNAs tested when compared to the wild-type cells. For *Rpc31* mutant, the differences were apparent and consistent in YPD grown cells. Since all the subsequent experiments in this paper were done with YPD-grown cells, we have repeated our RNA hybridization experiment using RNA extracted from cells grown in YPD. We have presented this additional data and rationale in our revised manuscript. As per Reviewer #2's suggestion, this new figure compares the level of reduction relative to the untreated wild-type (which is defined as 100% for the different genes). In addition, we have presented samples of the original hybridization blots, as well as signals from all the loading controls used (U4, U5, and U3). This is now presented as Supplemental Fig. 1

Except for the data shown in Figure 1, CPZ treatment is presented in this manuscript as equivalent to cell stress. It may elicit unique effects that are not observed upon other forms of stress. Therefore, it would be important not only to determine whether nutrient starvation or tunicamycin treatment generate the same effects on Pol III transcription (shown in Fig 1), but also to analyze whether loss of *hmt1* activity or R5,9A mutation of *RPC31* result in lower repression of tRNA expression under these forms of stress?

Author's Response: We agree with the reviewers that there is a possibility for cells to elicit a unique response with CPZ. The focus of this paper is to understand the role of Hmt1 in the transcription of tRNA genes in the context of both stress and non-stress conditions. Herein we have shown that under such CPZ treatment, *Rpc31* methylation is required for the proper interaction between RNA Pol III and repressor Maf1. Since the work from Willis group showed that Maf1 mediates repression of RNA Pol III in yeast under diverse stress conditions, including treatment with CPZ, tunicamycin, and nutrient limitation (Upadhyaya et al. 2002), we presume that the mechanism by which *Rpc31* methylation promotes the interaction between RNA Pol III and Maf1 (as a response to these treatments) is likely to be the same. However, we agree with the reviewer that the extent of changes in tRNA output may vary under different conditions. The experiments to address these concerns are interesting but we feel that they fall outside the scope of our current study.

Hmt1 also interacts with the B' subunit of TFIIB. Does this interaction have an influence on the observed effects on tRNA expression?

Author's Response: In our previous work, we have shown the interaction between Bdp1 (B'') with Hmt1 (Milliman et al. 2012). Our working model is that that Bdp1 recruits Hmt1 to the tRNA genes to allow for the methylation of *Rpc31*. As per Reviewer #1's suggestion, we have performed a co-immunoprecipitation assay to show that Hmt1 is able to be co-immunoprecipitated by *Rpc31*. This data is now presented as Fig. 5D in our revised manuscript. While the reviewer suggests an interesting question, we feel

this is outside our current scope of this paper and we will try to address this in our future studies.

Minor concerns :

The authors show in Figure 1 that the transcription of the 5S gene is not altered upon starvation, CPZ or tunicamycin treatment. Is the same true for deletion of hmt1 activity or RPC31 R5,9A mutation? Is the Pol III transcribed 5S gene differently affected by RPC31 methylation compared to tRNA genes?

Author's Response: Our Figure 1 depicts results from our ChIP data where we show a lack of significant change in the Hmt1 occupancy at 5S gene upon starvation, CPZ, or tunicamycin treatment. We do not have any data showing how RNA Pol III transcription on the 5S gene may be changed due to the loss of Hmt1 activity or Rpc31^{R5,9A} mutation.

The finding that RPC32 β , but not RPC32 α is modified by methylation is counterintuitive with respect to their supposed functions in cellular homeostasis and stem cell- or tumor-supporting activities. RPC31 methylation is associated with increased tRNA levels during normal or optimal growth and with protection from repression under cellular stress. Therefore, it does not make sense that the homeostasis-regulating subunit RPC32 β is associated with this tRNA level enhancing modification. It would make sense if the tumor-associated RPC32 α subunit would be methylated, thereby protecting tumor cells from reduced tRNA levels during stress and allowing high expression levels under normal growth conditions, thereby favoring tumor growth. However, the data on methylation of RPC31 and RPC32 β seem to be solid and I believe that the functional outcome of this modification needs to be better characterized for publication of these data. See also first comment in major concerns.

Author's Response: We have provided our thoughts below on why methylation potential is seen with RPC32 β , but not RPC32 α . In normal cell growth, RPC32 β is ubiquitously expressed but not RPC32 α . However, RPC32 α expression is increased in transformed cells (Haurie et al., 2010). Furthermore, MAF1 expression and activity inversely correlate with the oncogenic activity (Shor et al., 2010, and Pailan et al., 2014). Given all these observation, our working model is that methylation of RPC32 β makes this protein regulatable by MAF1 in normal cellular homeostasis. In transformed cells, however, a higher cellular demand for Pol III transcripts exists and to escape the intrinsic ability of MAF1 in repressing Pol III transcription, these transformed cell increase the overall pool of Rpc32 proteins with the alpha subunit version. In doing so, the intracellular pool of RPC32 β available for incorporating into RNA Pol III is diluted. Incorporating RPC32 α does not affect the ability of Pol III to be recruited its target genes since the Hernandez lab has shown that RNA Pol III incorporating either RPC32 α or RPC32 β are able to target the same set of genes (Renaud et al., 2014). However, incorporating RPC32 α instead of RPC32 β will make such Pol III no longer responsive to MAF1 repression. Thereby allowing for the cell to escape or bypass the negative regulation by MAF1. We have included this discussion in our revised manuscript.

The Wang & Roeder paper published in Genes & Development in 1997 is cited for the description of the yeast RPC31-RPC34-RPC82 complex. However, this paper describes the human RPC32-RPC39-RPC62 complex. A more appropriate citation would be Werner et al., 1992, also supported by the data published by Thuillier et al., 1995.

Author's Response: We have now included the references mentioned.

May 16, 2019

RE: Life Science Alliance Manuscript #LSA-2018-00261-TR

Dr. Michael C Yu
SUNY-Buffalo
Biological Sciences
109 Cooke Hall
Buffalo, NY 14051

Dear Dr. Yu,

Thank you for submitting your revised manuscript entitled "Robust Repression of tRNA Gene Transcription During Stress Requires Protein Arginine Methylation". As you will see, the reviewers appreciate the effort that went into the revision, though reviewer #1 points to a few remaining concerns. We would thus be happy to publish your paper in Life Science Alliance pending final revisions necessary to address reviewer #1's concerns. Furthermore, please upload the tables as word docx or excel files when submitting such a revised version. Some of the blots in figure 5 are heavily over-contrasted, it would be good to provide shorter exposure times as source data (see paragraph on source data in section B below).

A. FINAL FILES:

B. MANUSCRIPT ORGANIZATION AND FORMATTING:

Sincerely,

Reviewer #1 (Comments to the Authors (Required)):

My comments to the original version of the manuscript are sufficiently addressed experimentally, however, the main text requires some additional explanation and information.

Although the original hybridization blots with the appropriate loading controls are now presented in Fig. S1, the legend should provide more precise information concerning the growth conditions of cells used in this study.

In general, it should be stated in the text or figure legends how these original blots correspond to quantification data shown in main figures. In particular, it is unclear which blot shows transcripts in the wild type and *hmt1*Δ cells grown at 30°C on synthetic complete (SC) media plus glucose, the one which corresponds to the quantification shown in Fig. 2F (left panel)?

The upper panel of Fig. S1 does not support the conclusion that Hmt1 is required to promote transcription of tRNA genes since the difference between wild type and *hmt1* Δ is not visible. It is hard to believe that this blot was used for the quantification shown in revised Fig. 2F. Moreover, upper panel of the revised Fig. 3A shows no difference between untreated wild type and *hmt1*Δ cells which is inconsistent with left panel of Fig. 2F.

The results for *hmt1* mutant are more convincing; possibly the authors may eliminate part of 2F showing *hmt1* Δ data.

Concerning my last comment to the original version of the manuscript: the authors demonstrated the interaction between Hmt1 and Rpc31 as requested, but they do not comment previous result on the Bdp1-Hmt1 interaction.

As suggested before, an hypothetical effect of Bdp1-Hmt1 interaction on Rpc31 methylation should be discussed in the main text on the basis of the known structure of the Pol III initiation complex.

Reviewer #3 (Comments to the Authors (Required)):

The authors have appropriately addressed all the questions I raised in my review of the initially submitted manuscript. The data represent a clear advance in the knowledge about RNA polymerase III transcription and I now recommend publication in Life Science Alliance.

Response to Reviewer #1

1. Although the original hybridization blots with the appropriate loading controls are now presented in Fig. S1, the legend should provide more precise information concerning the growth conditions of cells used in this study.

In general, it should be stated in the text or figure legends how these original blots correspond to quantification data shown in main figures. In particular, it is unclear which blot shows transcripts in the wild type and *hmt1Δ* cells grown at 30°C on synthetic complete (SC) media plus glucose, the one which corresponds to the quantification shown in Fig. 2F (left panel)?

Author's Response: In our Materials and Methods section, we have stated that cells are all grown at 30°C on YPD except for the RNA hybridization for *hmt1Δ* cells in Fig. 2F. Nevertheless, we have added this information in the figure legends to further clarify any potential confusion. The sample original blots shown on Fig. S1 correspond to the data presented in Fig. 3, not Fig. 2F.

2. The upper panel of Fig. S1 does not support the conclusion that Hmt1 is required to promote transcription of tRNA genes since the difference between wild type and *hmt1Δ* is not visible. It is hard to believe that this blot was used for the quantification shown in revised Fig. 2F. Moreover, upper panel of the revised Fig. 3A shows no difference between untreated wild type and *hmt1Δ* cells which is inconsistent with left panel of Fig. 2F.

Author's Response: The blots shown in Fig. S1 were not the ones used for quantification in Fig. 2F. Rather, these blots were used for calculating the bar graphs in Figs. 3A and 3B. When grown in glucose, the differences seen with *hmt1Δ* cells is modest and not consistent, hence you do not see this much of a difference when the blot is subjected to quantification (as seen on the bar graphs as well). In this revised Fig. S1, we have enclosed the sample blots that were used for the quantification data shown in Fig. 2F (e.g. grown in SD+glucose). There is a clear difference between wild-type and *hmt1Δ* which reflects the changes seen in the bar graphs shown in Fig. 2F.

3. Concerning my last comment to the original version of the manuscript: the authors demonstrated the interaction between Hmt1 and Rpc31 as requested, but they do not comment previous result on the Bdp1-Hmt1 interaction.

As suggested before, an hypothetical effect of Bdp1-Hmt1 interaction on Rpc31 methylation should be discussed in the main text on the basis of the known structure of the Pol III initiation complex.

Author's Response: We have added a new section in our discussion to address Reviewer's suggestion in the main text.

May 22, 2019

RE: Life Science Alliance Manuscript #LSA-2018-00261-TRR

Dr. Michael C Yu
SUNY-Buffalo
Biological Sciences
109 Cooke Hall
Buffalo, NY 14051

Dear Dr. Yu,

Thank you for submitting your Research Article entitled "Robust Repression of tRNA Gene Transcription During Stress Requires Protein Arginine Methylation". It is a pleasure to let you know that your manuscript is now accepted for publication in Life Science Alliance. Congratulations on this interesting work.

DISTRIBUTION OF MATERIALS:

Again, congratulations on a very nice paper. I hope you found the review process to be constructive and are pleased with how the manuscript was handled editorially. We look forward to future exciting submissions from your lab.

Sincerely,
